# Coresets for Time Series Clustering

**Lingxiao Huang**
Tsinghua University

**K. Sudhir**
Yale University

**Nisheeth K. Vishnoi**
Yale University

## Abstract

We study the problem of constructing coresets for clustering problems with time series data. This problem has gained importance across many fields including biology, medicine, and economics due to the proliferation of sensors for real-time measurement and rapid drop in storage costs. In particular, we consider the setting where the time series data on $N$ entities is generated from a Gaussian mixture model with autocorrelations over $k$ clusters in $\mathbb{R}^d$. Our main contribution is an algorithm to construct coresets for the maximum likelihood objective for this mixture model. Our algorithm is efficient, and, under a mild assumption on the covariance matrices of the Gaussians, the size of the coreset is independent of the number of entities $N$ and the number of observations for each entity, and depends only polynomially on $k$, $d$ and $1/\varepsilon$, where $\varepsilon$ is the error parameter. We empirically assess the performance of our coresets with synthetic data.

## 1 Introduction

A multivariate time series dataset, represented as $\mathcal{X} = \left\{ X_i = (x_{i1}, \ldots, x_{i,T_i}) \subset \mathbb{R}^{T_i \times d} \mid i \in [N] \right\}$, where $N$ is the number of entities, $T_i$ is the number of time periods corresponding to entity $i$ and $d$ is the number of features, tracks features of a cross-section of entities longitudinally over time. Such data is also referred to as panel data [8] and has seen rapid growth due to proliferation of sensors, IOT and wearables that facilitate real time measurement of various features associated with entities and the rapid drop in storage costs. Specific examples of time series data include biomedical measurements (e.g., blood pressure and electrocardiogram), epidemiology and diffusion through social networks, weather, consumer search, content browsing and purchase behaviors, mobility through cell phone and GPS locations, stock prices and exchange rates in finance [2].

Computational problems of interest with time series data include pattern discovery [49], regression/prediction [39], forecasting [9], and clustering [47, 31, 2] which arises in applications such as anomaly detection [46]. Though clustering is a central and well-studied problem in unsupervised learning, most standard treatments of clustering tend to be on static data with one observation per entity [35, 11]. Time series clustering introduces a number of additional challenges and is an active area of research, with many types of clustering methods proposed [15]; direct methods on raw data, indirect methods based on features generated from the raw data, and model based clustering where the data are assumed to be generated from a model. For surveys of time series clustering, see [47, 2].

We focus on model-based time series clustering using a likelihood framework that naturally extends clustering with static data [11] to time series data [47], where each multivariate time series is generated by one of $k$ different parametrically specified models. For a survey of the importance of this sub-literature and its various real-world applications, see [30]. A prevalent approach is to assume a finite mixture of data generating models, with Gaussian mixture models being a common specification [50, 53]. Roughly, the problem is to partition $\mathcal{X}$ probabilistically into $k$ clusters, grouping those series

35th Conference on Neural Information Processing Systems (NeurIPS 2021).

generated by the same time series model into one cluster. Generically, the model-based $k$-clustering problem can be formulated as: $\arg\max_{\alpha, \theta^{(1)}, \ldots, \theta^{(k)}} \sum_{i \in [N]} \ln \sum_{l \in [k]} \alpha_l \cdot p_i(\mathcal{X} | \theta^{(l)})$, where $\alpha_l$ is the probability of a time series belonging to cluster $l$ and $p_i(\mathcal{X} | \theta^{(l)})$ is the likelihood of the data of entity $i$, given that the data is generated from model $l$ (with parameters $\theta^{(l)}$). Temporal relationships in time series are commonly modeled as autocorrelations (AR)/moving averages (MA) [20, 61] that account for correlations across observations, and hidden Markov models (HMM) where the underlying data generating process is allowed to switch periodically [21, 62]. This paper focuses on the case when the data generation model for each segment $l \in [k]$ is a multivariate Gaussian with autocorrelations. More formally, the generative model for each cluster is from a Gaussian mixture: $x_{it} := \mu^{(l)} + e_{it}$, where $e_{it} = N(0, \Sigma^{(l)}) + \Lambda^{(l)} e_{i,t-1}$, where $N(0, \Sigma^{(l)})$ captures the mixture of Gaussian distributions from which entity level observations are drawn, and $\Lambda$ captures the correlation between two successive observations through an AR(1) process [32, 44]. Overall, this model can be represented with cluster level model parameters $\theta^{(l)} = \{\mu^{(l)}, \Sigma^{(l)}, \Lambda^{(l)}\}$ and cluster probability $\alpha_l$.

Time series datasets are also much larger than static datasets [47]. For instance, as noted in [42], ECG (electrocardiogram) data requires 1 GB/hour, a typical weblog requires 5 GB/week. Such large sizes lead to high storage costs and also make it necessary to work with a subset of the data to conduct the clustering analysis to be computationally practical. Further, long time series data on entities also entails significant data security and privacy risks around the entity as it requires longer histories of entity behaviors to be stored and maintained [25, 41]. Thus, there is significant value for techniques that can produce approximately similar clustering results with smaller samples as with complete data.

Coresets have emerged as an effective tool to both speed up and reduce data storage by taking into account the objective and carefully sampling from the full dataset in a way that any algorithm run on the sampled set returns an approximate solution for the original objective with guaranteed bounds on approximation error [34]. Coresets have been developed for both unsupervised clustering (e.g., $k$-means, Gaussian mixture models) and supervised machine learning (e.g., regression) methods for static data; for surveys on coresets for static data, see [7, 22]. A natural question is whether coresets can be also useful for addressing the time series clustering problem. Recently, there is evidence that small coresets can be efficiently constructed for time series data, albeit for regressions with time series (panel) data [37]. However, as we explain in Section 2, there are challenges in extending coresets for static data clustering and time series (panel) data regressions for time series clustering.

**Our contributions**. We study coresets for a general class of time series clustering in which all entities are drawn from Gaussian mixtures with autocorrelations (as described above; see Problem 3.1). We first present a definition of coresets for the log-likelihood $k$-clustering objective for this mixture model. One issue is that this objective cannot be decomposed into a summation of entity-time objectives due to the interaction of the $\ln$ term from log-likelihood and the $\exp$ term from Gaussians (Problem 3.1). To estimate this objective using a coreset, we introduce an analogous clustering objective on a subset of data (Definition 3.3). Our main result is an algorithm to construct coresets for this objective for the aforementioned mixture model (Theorem 4.3). Our algorithm is efficient (linear on both $N$ and $\sum_{i \in [N]} T_i$) and, assuming that the condition number of the covariance matrices of the Gaussians ($\Sigma^{(l)}$) are bounded and the eigenvalues of the autocorrelation matrices ($\Lambda^{(l)}$) are bounded away from 1 (Assumption 4.1), the size of our coreset is independent of the number of entities $N$ and the number of observations $T_i$s; it only polynomially depends on $k$, $d$ and $1/\varepsilon$, where $\varepsilon$ is the error parameter (Theorem 4.3).

Our coreset construction (Algorithm 1) leverages the Feldman-Langberg (FL) framework [27]. Due to multiple observations for each entity and associated autocorrelations, the objective is non-convex and quite complicated in contrast to the static setting where the objective only contains one observation drawn from Gaussian mixtures for each entity. Thus, bounding the "sensitivity" (Lemma 4.6) of each entity, which is a key step in coreset construction using the FL-framework, becomes challenging. We handle this technical difficulty by 1) upper-bounding the maximum effect of covariances and autocorrelations by using the observation that the gap between the clustering objectives with and without $\Sigma$ and $\Lambda$ is always constant, and 2) reducing the Gaussian mixture time series clustering problem to a certain $k$-means clustering problem whose total sensitivity is upper bounded.

Empirically, we assess the performance of our coreset on synthetic data for a three cluster problem (Section 5), and compare the performance with two benchmarks: uniform sampling and a static coreset benchmark (**LFKF** [48]) in which we do not consider autocorrelations and regard time series

data as static. We find that our coreset performs better relative to uniform sampling and **LFKF** on both data storage and computation speed for a range of accuracy guarantees: To achieve a similar fit with the full data, our coreset needs fewer entity-time observations (<40%); the computation time for a given level of accuracy reduces by a 3x-22x factor when compared to uniform sampling and **LFKF**. Moreover, our coreset speeds up the computation time relative to the full data by 14x-171x. We note that the performance advantage of our coreset is greater when there is more time series data relative to entities ($T_i \gg N$).

## 2   Related work

Time series or panel data analysis are important in many fields– biological [18], engineering [3], economics [51] and the social sciences [59]. Clustering has been a central problem in unsupervised learning and also has a long history of use across many fields; for a historical overview of applications across fields, see [12]. While most standard treatments of clustering in machine learning tend to be on static data [35, 11], there is by now a significant and rapidly growing literature on time series clustering; recent surveys include [47, 2]. While there are many coresets for algorithms on static data (including for clustering), there has been little work on coresets on time series data (see below).

**Coresets for clustering.** Coresets for various clustering objectives on static data have been well studied, including $k$-median [34, 17, 55, 38, 19], $k$-means [34, 17, 28, 13, 10, 55, 38, 19], and $k$-center [1, 33]. For surveys of static coresets, see [52, 23]. [48, 26] present coreset constructions for clustering problems on static data generated using Gaussian mixture models. [48] constructs a coreset of size poly$(k, d, 1/\varepsilon)$ under the same boundedness assumption as ours. [26] remove this assumption and construct a coreset whose size additionally depends on poly $\log N$, but require another assumption that all $x_i$s are integral and in a bounded range. In contrast, we consider a generalized problem where each entity has multiple observations over time and accounts for autocorrelations across successive observations. The new idea is to reduce the GMM time series clustering problem (Problem 3.1) to a $k$-means clustering problem whose points are linear combinations of those in $X_i$ (Lemma A.12), and show that the effect of autocorrelation parameters $\Lambda$ on the entity objectives can be upper bounded (Lemma A.11).

**Regression with time series data.** [37] construct coresets for a regression problem, called GLSE$_k$, for time-series data including autocorrelations and a $k$-partition of entities. This is a supervised learning task that includes an additional label $y_{it}$ for each entity-time pair. It aims to find a $k$-partition of entities and additional regression vectors $\beta^{(l)} \in \mathbb{R}^d$ for each partition $l \in [k]$ such that a certain linear combination of terms $\|y_{it} - (\beta^{(l)})^\top x_{it}\|_2^2$ is minimized. In contrast, our problem is an unsupervised learning task, estimates points $x_{it}$ by Gaussian means $\mu$ instead of $\beta$, and includes covariances $\Sigma$ between different features. The definitions of coresets in [37] and ours consist of entity-time pairs but our coresets need to include multiple weight functions ($w$ for entities and $w^{(i)}$ for time periods of selected entity $i$; see Definition 3.3) instead of one weight function as in the regression problem. This is to deal with the interaction of the ln term from the log-likelihood and the exp term from the Gaussians. [37]'s result relies on an assumption that the regression objective of each entity is in a bounded range over the parameter space. This may not be satisfied in our setting since the clustering objective of an entity may be unbounded when all Gaussian means $\mu^{(l)}$ are far away from observations of this entity. To bypass this, we reduce our problem to a $k$-means clustering problem (Definition 4.2), whose total sensitivity is provably $O(k)$ (Lemma A.12), by upper bounding the affect of covariance and autocorrelation parameters on the clustering objectives of entities (Lemma A.11) under the assumption that the condition number of covariance matrix is bounded (Assumption 4.1), which provides an upper bound for the total sensitivity of entities on our problem (Lemma 4.6).

Another related direction is to consider coreset construction for $k$-segmentation with time series data [54, 29], which aims to estimate the trace of an entity by a $k$-piecewise linear function ($k$-segment). Note that the case of $k = 1$ is equivalent to the linear regression problem. [54, 29] proposed efficient coreset construction algorithms which accelerate the computation time of the $k$-segmentation problem. The main difference from our setting is that they consider a single entity observed at multiple time periods, and the objective is an additive function on all time periods. This enables them to relate the $k$-segmentation problem to the static setting. It is interesting to investigate coreset construction for more optimization problems with time series data.

# 3 Clustering model and coreset definition

Given $\mathcal{X} = \{X_i = (x_{i,1}, \ldots, x_{i,T_i}) \in \mathbb{R}^{d \times T_i} \mid i \in [N]\}$, we first model a general class of time series clustering problems. Then we specify our setting with Gaussian mixture time series data (Problem 3.1), and define coresets accordingly (Definition 3.3).

**Clustering with time series data.** Given an integer $k \geq 1$, let $\Delta_k \subset \mathbb{R}^k$ denote a probability simplex satisfying that for any $\alpha \in \Delta_k$, we have $\alpha_i \in [0, 1]$ for $i \in [k]$ and $\sum_{l \in [k]} \alpha_i = 1$. Let $\mathcal{P}$ denote a parameter space where each $\theta \in \mathcal{P}$ represents a specific generative model of time series data. For each entity $i \in [N]$ and model $\theta \in \mathcal{P}$, define $p_i(\mathcal{X} \mid \theta) := \Pr[X_i \mid \theta]^{1/T_i}$ to be the average likelihood/realized probability of $X_i$ from model $\theta$. The ratio $1/T_i$ is used to normalize the contribution to the objective of entity $i$ due to different lengths $T_i$. The time series clustering problem is to compute $\alpha \in \Delta_k$ and $\theta = (\theta^{(1)}, \ldots, \theta^{(k)}) \in \mathcal{P}^k$ that minimizes the negative log-likelihood, i.e., $-\sum_{i \in [N]} \ln \sum_{l \in [k]} \alpha_l \cdot p_i(\mathcal{X} \mid \theta^{(l)})$. Here, for each $l \in [k]$, $\alpha_l$ represents the probability that each time series is generated from model $\theta^{(l)}$. Note that the clustering objective depends on the choice of model family $\mathcal{P}$. In this paper, we consider the following specific model family $\mathcal{P}$ that each time series is generated from Gaussian mixtures.

**GMM clustering with time series data.** Let $\alpha \in \Delta_k$ be a given probability vector. Let $\mathcal{S}^d$ denote the collection of all symmetric positive definite matrices in $\mathbb{R}^{d \times d}$. For each $i \in [N]$, with probability $\alpha_l$ ($l \in [k]$), let $x_{it} := \mu^{(l)} + e_{it}$ for each $t \in [T_i]$ where $\mu^{(l)} \in \mathbb{R}^d$ represents a Gaussian mean and $e_{it} \in \mathbb{R}^d$ represents the error vector drawn from the following normal distribution: $e_{it} := \Lambda^{(l)} e_{i,t-1} + N(0, \Sigma^{(l)})$, where $\Lambda^{(l)} \in \mathcal{S}^d$ is an AR(1) autocorrelation matrix and $\Sigma^{(l)} \in \mathcal{S}^d$ is the covariance matrix of a multivariate Gaussian distribution. Now we let $\mathcal{P} := \mathbb{R}^d \times \mathcal{S}^d \times \mathcal{S}^d$ and note that each Gaussian generative model can be represented by a tuple $\theta = (\mu, \Sigma, \Lambda) \in \mathcal{P}$. Moreover, we have that the realized probability of each entity $i \in [N]$ is

$$p_i(\mathcal{X} \mid \theta) = p_i(\mathcal{X} \mid \mu, \Sigma, \Lambda) := \frac{\exp(-\frac{1}{2T_i} \psi_i(\mu, \Sigma, \Lambda))}{(2\pi)^{d/2} |\Sigma|^{1/2}}, \tag{1}$$

where $|\Sigma|$ is the determinant of $\Sigma$ and $\psi_i(\mu, \Sigma, \Lambda) := \sum_{t \in [T_i]} \psi_{it}(\Sigma, \Lambda)$ with

$$\psi_{i1}(\Sigma, \Lambda) := (x_{i,1} - \mu)^\top \Sigma^{-1}(x_{i,1} - \mu) - (\Lambda(x_{i,1} - \mu))^\top \Sigma^{-1} (\Lambda(x_{i,1} - \mu)), \text{ and}$$

$$\psi_{it}(\Sigma, \Lambda) := ((x_{it} - \mu) - \Lambda(x_{i,t-1} - \mu))^\top \Sigma^{-1} ((x_{it} - \mu) - \Lambda(x_{i,t-1} - \mu)), \forall \ 2 \leq t \leq T_i.$$

We note that each sub-function $\psi_{it}$ contains at most two entity-time pairs: $x_{it}$ and $x_{i,t-1}$. Our Gaussian mixture time series model gives rise in the following clustering problem.

**Problem 3.1** (**Clustering with GMM time series data**). *Given a time series dataset $\mathcal{X} = \{X_i = (x_{i,1}, \ldots, x_{i,T_i}) \in \mathbb{R}^{d \times T_i} \mid i \in [N]\}$ and integer $k \geq 1$, the GMM time series clustering problem is to compute $\alpha \in \Delta_k$ and $\theta = (\mu^{(l)}, \Sigma^{(l)}, \Lambda^{(l)})_{l \in [k]} \in \mathcal{P}^k$ that minimize*

$$f(\alpha, \theta) := \sum_{i \in [N]} f_i(\alpha, \theta) = -\sum_{i \in [N]} \ln \sum_{l \in [k]} \alpha_l \cdot p_i(\mathcal{X} \mid \mu^{(l)}, \Sigma^{(l)}, \Lambda^{(l)}).$$

From Equation (1) it follows that the coefficient of each Gaussian before the $\exp$ operation is $\frac{\alpha_l}{(2\pi)^{d/2} |\Sigma^{(l)}|^{1/2}}$, whose summation $Z(\alpha, \theta) := \sum_{l \in [k]} \frac{\alpha_l}{(2\pi)^{d/2} |\Sigma^{(l)}|^{1/2}}$ is not a fixed value and depends on $\Sigma^{(l)}$s. To remove this dependence on the coefficients, we define $\alpha_l'(\theta) := \frac{\alpha_l}{(2\pi)^{d/2} |\Sigma^{(l)}|^{1/2} Z(\alpha, \theta)}$ to be the normalized coefficient for $l \in [k]$, define offset function $\phi : \Delta_k \times \mathcal{P}^k \to \mathbb{R}$ to be $\phi(\alpha, \theta) := -N \ln Z(\alpha, \theta)$, and define $f'(\alpha, \theta) := -\sum_{i \in [N]} \ln \sum_{l \in [k]} \alpha_l \cdot \exp(-\frac{1}{2T_i} \psi_i(\mu^{(l)}, \Sigma^{(l)}, \Lambda^{(l)}))$ whose summation of coefficients before the $\exp$ operations is 1. This idea of introducing $f'$ also appears in [24, 26], which is useful for coreset construction. This leads to the following observation.

**Observation 3.2.** *For any $\alpha \in \Delta_k$ and $\theta \in \mathcal{P}^k$, $f(\alpha, \theta) = f'(\alpha'(\theta), \theta) + \phi(\alpha, \theta)$.*

**Our coreset definition.** The clustering objective $f$ can only be decomposed into the summation of $f_i$s instead of sub-functions w.r.t. entity-time pairs. A simple idea is to select a collection of weighted entities as a coreset. However, in doing so, the coreset size will depend on $T_i$s and fail to be independent of both $N$ and $T_i$s. To address this problem, we define our coreset as a collection of weighted entity-time pairs, which is similar to [37]. Let $P_\mathcal{X} := \{(i, t) \mid i \in [N], t \in [T_i]\}$ denote the collection of indices of all $x_{it}$. Given $S \subseteq P_\mathcal{X}$, we let $I_S := \{i \in [N] \mid \exists t \in [T_i], s.t., (i, t) \in S\}$ denote the collection of entities that appear in $S$. Moreover, for each $i \in I_S$, we let $J_{S,i} := \{t \in [T_i] : (i, t) \in S\}$ denote the collection of observations for entity $i$ in $S$.

**Definition 3.3** (**Coresets for GMM time series clustering**). *Given a time series dataset $\mathcal{X} = \{X_i = (x_{i,1}, \ldots, x_{i,T_i}) \in \mathbb{R}^{d \times T_i} \mid i \in [N]\}$, constant $\varepsilon \in (0,1)$, integer $k \geq 1$, and parameter space $\Delta_k \times \mathcal{P}^k$, an $\varepsilon$-coreset for GMM time series clustering is a weighted set $S \subseteq P_{\mathcal{X}}$ together with weight functions $w : I_S \to \mathbb{R}_{\geq 0}$ and $w^{(i)} : J_{S,i} \to \mathbb{R}_{\geq 0}$ for $i \in I_S$ such that for any $\alpha \in \Delta_k$ and $\theta = (\mu^{(l)}, \Sigma^{(l)}, \Lambda^{(l)})_{l \in [k]} \in \mathcal{P}^k$, $f'_S(\alpha, \theta) := -\sum_{i \in I_S} w(i) \cdot \ln \sum_{l \in [k]} \alpha_l \cdot \exp(-\frac{1}{2T_i} \sum_{t \in J_{S,i}} w^{(i)}(t) \cdot \psi_{it}(\mu^{(l)}, \Sigma^{(l)}), \Lambda^{(l)})) \in (1 \pm \varepsilon) \cdot f'(\alpha, \theta)$.*

Combining the above definition with Observation 3.2, we note that for any $\alpha \in \Delta_k$ and $\theta \in \mathcal{P}^k$,

$$f'_S(\alpha'(\theta), \theta) + \phi(\alpha, \theta) \in (1 \pm \varepsilon) \cdot f'(\alpha, \theta) + \phi(\alpha, \theta) \in (1 \pm \varepsilon) \cdot f(\alpha, \theta) \pm 2\varepsilon \phi(\alpha, \theta). \quad (2)$$

As $\varepsilon$ tends to 0, $f'_S(\alpha'(\theta), \theta) + \phi(\alpha, \theta)$ converges to $f(\alpha, \theta)$. Moreover, given such a coreset $S$, if we additionally have that $\phi(\alpha, \theta) \lesssim f(\alpha, \theta)$, we can minimize $f'_S(\alpha'(\theta), \theta) + \phi(\alpha, \theta)$ to solve Problem 3.1. Thus, if $\phi(\alpha, \theta) \lesssim f(\alpha, \theta)$, we conclude that $f'_S(\alpha'(\theta), \theta) + \phi(\alpha, \theta) \in (1 \pm 3\varepsilon) \cdot f(\alpha, \theta)$. Note that we introduce multiple weight functions ($w$ for entities and $w^{(i)}$ for time periods of selected entity $i$) in Definition 3.3 unlike the coreset for time series regression in [37] which uses only one weight function. This is because $f'_i$ contains $\ln$ and $\exp$ operators instead of linear combinations of $\psi_{it}$; it is unclear how to use a single weight function to capture both the entity and time levels.

## 4  Theoretical results

We present our coreset algorithm (Algorithm 1) and the main result on its performance (Theorem 4.3). Theorem 4.3 needs the following assumptions on covariance and autocorrelation parameters.

**Assumption 4.1.** *Assume (1) that there exists constant $D \geq 1$ such that $\frac{\max_{l \in [k]} \lambda_{\max}(\Sigma^{(l)})}{\min_{l \in [k]} \lambda_{\min}(\Sigma^{(l)})} \leq D$ where $\lambda_{\max}(\cdot)$ represents the largest eigenvalue and $\lambda_{\min}(\cdot)$ represents the smallest eigenvalue, and (2) for $l \in [k]$, $\Lambda^{(l)} \in \mathcal{D}_\lambda^d$ for some constant $\lambda \in (0,1)$. Here $\mathcal{D}_\lambda^d$ is a collection of all diagonal matrix in $\mathbb{R}^{d \times d}$ whose diagonal elements are at most $1 - \sqrt{\lambda}$.*

The first assumption requires that the condition number of each covariance matrix is upper-bounded, which also appears in [60, 45, 48] that consider Gaussian mixture models with static data. The second assumption, roughly, requires that there exist autocorrelations only between the same features. The upper bound for diagonal elements ensures that the autocorrelation degrades as time period increases, which is also assumed in [37]. Note that both the eigenvalues of $\Sigma^{(i)}$ and the positions of means $\mu^{(l)}$s affect cost function $\psi_i$s. For instance, consider the case that all eigenvalues are 1 and all autocorrelations are 0, i.e., for all $l \in [k]$, $\Sigma^{(l)} = I_d$ and $\Lambda^{(l)} = 0_d$. In this case, it is easy to see that $\frac{\max_{\mu \in \mathcal{R}^d} \psi_i(\mu, I_d, 0_d)}{\min_{\mu \in \mathcal{R}^d} \psi_i(\mu, I_d, 0_d)} = \infty$, i.e., the value of $\psi_i(\mu, I_d, 0_d)$ is unbounded as $\mu$ changes. Differently, the component's cost functions are bounded in [37] since $\mu^{(l)}$s do not appear in [37] that consider regression problems.

Let $\mathcal{P}_\lambda^k := (\mathbb{R}^d \times \mathcal{S}^d \times \mathcal{D}_\lambda^d)^k$ denote the parameter space. For preparation, we propose the following $k$-means clustering problem.

**Definition 4.2** ($k$**-means clustering of entities**). *Given an integer $k \geq 1$, the goal of the $k$-means clustering problem of entities of $X$ is to find a set $C^\star = \{c_1^\star, \ldots, c_k^\star\} \subset \mathbb{R}^d$ of $k$ centers that minimizes $\sum_{i \in [N]} \min_{l \in [k]} \|\frac{\sum_{t \in [T_i]} x_{it}}{T_i} - c_l\|_2^2$ over all $k$ center sets $C = \{c_1, \ldots, c_k\} \subset \mathbb{R}^d$. Let $\mathsf{OPT}^{(O)}$ denote the optimal $k$-means value of $C^\star$.*

Note that there exists an $O(2^{\mathsf{poly}(k)} N d)$ time algorithm to compute a nearly optimal solution for both $\mathsf{OPT}^{(O)}$ and $C^\star$ [43]. Another widely used algorithm is called $k$-means++ [6], which provides an $O(\ln k)$-approximation in $O(N d k \ln N \ln k)$ time but performs well in practice. We also observe that $\psi_i(\mu, I_d, 0_d) = \|\frac{\sum_{t \in [T_i]} x_{it}}{T_i} - \mu\|_2^2 + \frac{1}{T_i} \sum_{t \in [T_i]} \|x_{it}\|_2^2 - \frac{\|\sum_{t \in [T_i]} x_{it}\|_2^2}{T_i^2}$ for any $\mu \in \mathbb{R}^d$. This observation motivates us to consider a reduction from Problem 3.1 to Definition 4.2, which is useful for constructing $I_S$.

**Our coreset algorithm.** We first give a summary of our algorithm (Algorithm 1). The input to Algorithm 1 is a time series dataset $\mathcal{X}$, an error parameter of coreset $\varepsilon \in (0,1)$, an integer $k \geq 1$ of

clusters, $\lambda \in (0, 1)$ (vector norm bound), and a covariance eigenvalue gap $D \geq 1$. We develop a two-staged importance sampling framework which first samples a subset $I_S$ of entities (Lines 1-8) and then samples a subset of time periods $J_{S,i}$ for each selected entity $i \in I_S$ (Lines 9-14).

In the first stage, we first set $M$ as the number of selected entities $|I_S|$ (Line 1). Then we solve the $k$-means clustering problem over entity means $b_i$ (Definition 4.2), e.g., by $k$-means++ [6] (Lines 2-3), and obtain an (approximate) optimal clustering value $\mathsf{OPT}^{(O)}$, a $k$-center set $C^\star := \{c_1^\star, \ldots, c_k^\star\} \subset \mathbb{R}^d$, and an offset value $a_i$ for each $i \in [N]$. Next, based on the distances of points $b_i$ to $C^\star$, we partition $[N]$ into $k$ clusters where entity $i$ belongs to cluster $c_{p(i)}^\star$ (Line 4). Based on $b_i$, $c_{p(i)}^\star$ and $\mathsf{OPT}^{(O)}$, we compute an upper bound $s(i)$ for the sensitivity w.r.t. entity $i \in [N]$ (Lines 5-6). Finally, we sample a weighted subset $I_S$ of entities as our coreset for the entity level by importance sampling based on $s(i)$ (Lines 7-8), which follows from the Feldman-Langberg framework (Theorem A.6).

In the second stage, we first set $L$ as the number of time periods $|J_{S,i}|$ for each selected entity $i \in I_S$ (Line 9). Then for each selected entity $i \in I_S$, we compute $\mathsf{OPT}_i^{(O)}$ as the optimal 1-means clustering objective of $x_{it}$s (Line 10). Next, we compute an upper bound $s_i(t)$ for the sensitivity w.r.t. time period $t \in [T_i]$ (Lines 11-12) based on $b_i$ and $\mathsf{OPT}_i^{(O)}$. Finally, we sample a weighted subset $J_{S,i}$ of time periods by importance sampling based on $s_i(t)$ (Lines 13-14).

---

**Algorithm 1:** CRGMM: Coreset construction for GMM time series clustering

**Input:** $\mathcal{X} = \left\{ X_i = (x_{i,1}, \ldots, x_{i,T_i}) \in \mathbb{R}^{d \times T_i} \mid i \in [N] \right\}$, constant $\varepsilon, \lambda \in (0, 1)$, number of clusters $k \geq 1$, dimension of autocorrelation vectors $q \geq 0$, constant of the variance gap $D \geq 1$, and parameter space $\mathcal{P}_\lambda^k = (\mathbb{R}^d \times \mathcal{S}^d \times \mathcal{D}_\lambda^d)^k$.

**Output:** a subset $S \subseteq P$ together with weight functions $w : I_S \to \mathbb{R}_{\geq 0}$ and $w^{(i)} : J_{S,i} \to \mathbb{R}_{\geq 0}$ for $i \in I_S$.

% Constructing a subset of entities

1: $M \leftarrow O\left( \frac{kD}{\lambda \varepsilon^2} (k^4 d^4 + k^3 d^8) \ln \frac{k}{\lambda} \right)$.

2: For each $i \in [N]$, compute point $b_i \in \mathbb{R}^d$ by $b_i \leftarrow \frac{\sum_{t \in [T_i]} x_{it}}{T_i}$ and value $a_i \geq 0$ by

$a_i \leftarrow \frac{1}{T_i} \sum_{t \in [T_i]} \|x_{it}\|_2^2 - \frac{\|\sum_{t \in [T_i]} x_{it}\|_2^2}{T_i^2}$. Let $A \leftarrow \sum_{i \in [N]} a_i$.

3: Compute (an $O(1)$-approximate for) $\mathsf{OPT}^{(O)}$ and $C^\star = \{c_1^\star, \ldots, c_k^\star\} \subset \mathbb{R}^d$ for the $k$-means clustering problem over points $b_i$ (Definition 4.2),[1] by [43, 6].

4: For each $i \in [N]$, compute index $p(i) \in [k]$ by $p(i) \leftarrow \arg\min_{l \in [k]} \|b_i - c_l^\star\|_2^2$.

5: For each $i \in [N]$, $s^c(i) \leftarrow \frac{1}{\left| \left\{ i' \in [N] : p(i') = c_{p(i)}^\star \right\} \right|}$.

6: For each $i \in [N]$, $s(i) \leftarrow \min \left\{ 1, 4D \left( \frac{4\|b_i - c_{p(i)}^\star\|_2^2}{\mathsf{OPT}^{(O)} + A} + 3s^c(i) \right) / \lambda \right\}$.

7: Pick a random sample $I_S \subseteq [N]$ of size $M$, where each $i \in I_S$ is selected w.p. $\frac{s(i)}{\sum_{i' \in [N]} s(i')}$

8: For each $i \in I_S$, $w(i) \leftarrow \frac{\sum_{i' \in [N]} s(i')}{M \cdot s(i)}$.

% Constructing a subset of time periods for each selected entity

9: $L \leftarrow O\left( \frac{Dd^8 \ln \frac{1}{\lambda}}{\lambda \varepsilon^2} \right)$.

10: For each $i \in I_S$, compute $\mathsf{OPT}_i^{(O)} \leftarrow \sum_{t \in [T_i]} \|x_{it} - b_i\|_2^2$.

11: For each $i \in I_S$ and $t \in [T_i]$, $s_i^c(t) \leftarrow \frac{2\|x_{it} - b_i\|_2^2}{\mathsf{OPT}_i^{(O)}} + \frac{6}{T_i}$.

12: For each $i \in I_S$ and $t \in [T_i]$, $s_i(t) \leftarrow \min \left\{ 1, 4D\lambda^{-1} \left( s_i^c(t) + \sum_{j=1}^{\min\{t-1,1\}} s_i^c(t - j) \right) \right\}$.

13: For each $i \in I_S$, pick a random sample $J_{S,i}$ of $L$ points, where each $t \in J_{S,i}$ is selected w.p.
$\frac{s_i(t)}{\sum_{t' \in [T_i]} s_i(t')}$.

14: For each $i \in I_S$ and $t \in J_{S,i}$, $w^{(i)}(t) \leftarrow \frac{\sum_{t' \in [T_i]} s_i(t')}{L \cdot s_i(t)}$.

% Output coreset $S$ of entity-time pairs

15: Let $S \leftarrow \{(i, t) \in P_\mathcal{X} : i \in I_S, t \in J_{S,i}\}$.

16: Output $\left( S, w, \left\{ w^{(i)} \right\}_{i \in I_S} \right)$.

---

[1]Here, we slightly abuse the notation by using $\mathsf{OPT}^{(O)}$ and $C^\star$ to represent the obtained approximate solution.

**Our main theorem.** Our main theorem is as follows, which indicates that Algorithm 1 provides an efficient coreset construction algorithm for GMM time series clustering.

**Theorem 4.3** (**Main result**). *Under Assumption 4.1, with probability at least* $0.9$*, Algorithm 1 outputs an* $\varepsilon$*-coreset of size* $O\left(\frac{D^2 k^5 d^{16} \ln^2 \frac{k}{\lambda}}{\lambda^2 \varepsilon^4}\right)$ *for GMM time-series clustering in* $O(d \sum_{i\in[N]} T_i + Ndk \ln N \ln k)$ *time.*

Note that the success probability in the theorem can be made $1 - \delta$ for any $\delta \in (0,1)$ at the expense of an additional factor $\log^2 1/\delta$ in the coreset size. The coreset size guaranteed by Theorem 4.3 has a polynomial dependence on factors $k, d, 1/\varepsilon, D$, and $1/\lambda$ and, importantly, does not depend on $N$ or $T_i$s. Compared to the clustering problem with GMM static data [48], the coreset has an additional dependence on $1/\lambda$ due to autocorrelation parameters. Compared to the regression problem (GLSE$_k$) with time series data [37, Theorem 5.2], the coreset does not contain the factor "$M$" that upper bounds the gap between the maximum and the minimum entity objective. The construction time linearly depends on the total number of observations $\sum_{i\in[N]} T_i$, which is efficient. The proof of Theorem 4.3 can be found in Section C.

**Proof overview of Theorem 4.3.** The proof consists of three parts: 1) Bounding the size of the coreset, 2) proving the approximation guarantee of the coreset, and 3) bounding the running time.

*Parts 1 & 2: Coreset size and correctness guarantee of Theorem 4.3.* We first note that the coreset size in Theorem 4.3 is $|S| = ML$. Here, $M$ is the number of sampled entities $|I_S|$ that guarantee that $I_S$ is a coreset for $f'$ at the entity level (Lemma 4.4). $L$ is the number of sampled observations $|J_{S,i}|$ for each $i \in I_S$ that guarantees that $J_{S,i}$ is a coreset for $\psi_i$ at the time level (Lemma 4.5).

**Lemma 4.4** (**The 1st stage outputs an entity-level coreset**). *With probability at least* $0.95$*, the output* $I_S$ *of Algorithm 1 with* $M := |I_S| = O\left(\frac{kD}{\lambda\varepsilon^2}(k^4 d^4 + k^3 d^8) \ln \frac{k}{\lambda}\right)$ *satisfies that for any* $\alpha \in \Delta_k$ *and* $\theta \in \mathcal{P}_\lambda^k$*,* $-\sum_{i\in I_S} w(i) \cdot \ln \sum_{l\in[k]} \alpha_l \cdot \exp(-\frac{1}{2T_i}\psi_i(\mu^{(l)}, \Sigma^{(l)}, \Lambda^{(l)})) \in (1\pm\varepsilon)\cdot f'(\alpha, \theta)$.

**Lemma 4.5** (**The 2nd stage outputs time level coresets for each** $i \in I_S$). *With probability at least* $0.95$*, for all* $i \in I_S$*, the output* $J_{S,i}$ *of Algorithm 1 with* $L := |J_{S,i}| = O\left(\frac{Dd^8 \ln \frac{1}{\lambda}}{\lambda\varepsilon^2}\right)$ *satisfies that for any* $(\mu, \Sigma, \Lambda) \in \mathcal{P}_\lambda$*,* $\sum_{t\in J_{S,i}} w^{(i)}(t) \cdot \psi_{it}(\mu, \Sigma, \Lambda) \in (1 \pm \varepsilon) \cdot \psi_i(\mu, \Sigma, \Lambda)$.

From Lemmas 4.4 and 4.5, we can conclude that $S$ is an $O(\varepsilon)$-coreset for GMM time-series clustering, since the errors $\varepsilon$ for the entity level and the time level are additively accumulated, and that the size of the coreset $ML$ as claimed; see Section C for a complete proof.

Both lemmas rely on the Feldman-Langberg framework [27, 13] (Theorem A.6). The key is to upper bound both the "pseudo-dimension" $\dim$ (Definition A.5) that measures the complexity of the parameter space, and the total sensitivity $\mathcal{G}$ (Definition A.4) that measures the sum of maximum influences of all sub-functions ($f'_i$ for Lemma 4.4 and $\psi_{it}$ for Lemma 4.5). The coreset size guaranteed by the Feldman-Langberg framework then is $O(\varepsilon^{-2}\mathcal{G} \dim \ln \mathcal{G})$.

For Lemma 4.5, we can upper bound the pseudo-dimension by $\dim = O(d^8)$ (Lemma B.1). For the total sensitivity, a key step is to verify that $s_i$ (Line 12) is a sensitivity function of $\psi_i$. This step uses a similar idea as in [37] and leads to showing a $\mathcal{G} = O(D/\lambda)$ bound for the total sensitivity (Lemma B.2). Then we verify that $L = O(\varepsilon^{-2}\mathcal{G} \dim \ln \mathcal{G}) = O\left(\frac{Dd^8 \ln \frac{1}{\lambda}}{\lambda\varepsilon^2}\right)$ is enough for Lemma 4.5 by the Feldman-Langberg framework (Theorem A.6). The proof can be found in Section B.

Lemma 4.4 is the most technical. The proof can be found in Section A and we provide a proof sketch.

*Proof sketch of Lemma 4.4.* By [4, 58], the pseudo-dimension is determined by the number of parameters that is upper bounded by $O(kd^2)$ and the number of operations on parameters that is upper bounded by $O(kd^6)$ including $O(k)$ exponential functions. Then we can upper bound the pseudo-dimension by $\dim = O(k^4 d^4 + k^3 d^8)$ (Lemma A.9 in Section A.2).

For the total sensitivity, the key is to verify that $s$ (Line 6) is a sensitivity function w.r.t. $f'$; this is established by Lemma 4.6 below. Then the total sensitivity is at most $\mathcal{G} = (16D + 12Dk)/\lambda$. This completes the proof of Lemma 4.4 since $M = O(\varepsilon^{-2}\mathcal{G} \dim \ln \mathcal{G}) = O\left(\frac{kD}{\lambda\varepsilon^2}(k^4 d^4 + k^3 d^8) \ln \frac{k}{\lambda}\right)$ is enough by the Feldman-Langberg framework (Theorem A.6).

**Lemma 4.6** (**$s$ is a sensitivity function w.r.t.** $f'$). *For each* $i \in [N]$*, we have* $s(i) \geq \max_{\alpha\in\Delta_k, \theta\in\mathcal{P}_\lambda^k} \frac{f'_i(\alpha,\theta)}{f'(\alpha,\theta)}$*. Moreover,* $\sum_{i\in[N]} s(i) \leq (16D + 12Dk)/\lambda$.

*Proof sketch of Lemma 4.6.* We first define $\psi_i^{(O)}(\mu) := \sum_{t \in [T_i]} \|x_{it} - \mu\|_2^2$ for any $\mu \in \mathbb{R}^d$, and define $f_i^{(O)}(\alpha, \theta^{(O)}) = -\ln \sum_{l \in [k]} \alpha_l \cdot \exp\left(-\frac{1}{2T_i \cdot \min_{l' \in [k]} \lambda_{\min}(\Sigma^{(l')})} \psi_i^{(O)}(\mu^{(l)})\right)$. Based on $f_i^{(O)}$, we define another sensitivity function $s^{(O)}(i) := \max_{\alpha \in \Delta_k, \theta^{(O)} \in \mathbb{R}^{d \times k}} \frac{f_i^{(O)}(\alpha, \theta^{(O)})}{f^{(O)}(\alpha, \theta^{(O)})}$ and want to reduce $s$ to $s_i^{(O)}$. Note that $\psi_i^{(O)}(\mu) = \psi_i(\mu, I_d, 0_d)$, i.e., $\psi_i^{(O)}$ removes the covariance and autocorrelation parameters from $\psi_i$. Due to this observation, we have that $s^{(O)}(i) \leq \max_{\alpha \in \Delta_k, \theta \in \mathcal{P}_\lambda^k} \frac{f_i'(\alpha, \theta)}{f'(\alpha, \theta)} \leq 4D \cdot s^{(O)}(i)/\lambda$ by upper bounding the affect of covariance parameters (controlled by factor $D$) and autocorrelation parameters (controlled by factor $1/\lambda$), summarized as Lemma A.11. Then to prove Lemma 4.6, it suffices to prove that $s^{(O)}(i) \leq \frac{4\|b_i - c_{p(i)}^\star\|_2^2}{\mathsf{OPT}^{(O)} + A} + 3s^c(i)$ which implies that the total sensitivity of $s^{(O)}$ is upper bounded by $4 + 3k$ (Lemma A.12), due to fact that $\sum_{i \in [N]} s^c(i) \leq k$ (Lemma A.13). Finally, the proof of Lemma A.12 is based on a reduction from $\psi^{(O)}$ to the $k$-means clustering problem of entities (Definition 4.2) by rewriting $\frac{1}{T_i} \psi_i^{(O)}(\mu) = \|b_i - \mu\|_2^2 + a_i$ and then projecting each $b_i$ to its closest center in $C^\star$. The full proof of this lemma can be found in Section A.3.

*Part 3: Running time in Theorem 4.3.* The first stage costs $O(d \sum_{i \in [N]} T_i + Ndk \ln N \ln k)$ time. The dominating steps are 1) to compute $b_i$ in Line 2 which costs $O(d \sum_{i \in [N]} T_i)$ time; 2) to solve the $k$-means clustering problem of entities (Line 3) which costs $Ndk \ln N \ln k)$ time. The second stage costs at most $O(d \sum_{i \in [N]} T_i)$ time. The dominating step is to compute $\mathsf{OPT}_i^{(O)}$ for all $i \in I_S$ in Line 10, where it costs $O(dT_i)$ time to compute each $\mathsf{OPT}_i^{(O)}$.

**Remark 4.7** (**Technical comparison with prior works**). *Note that our entity coreset construction uses some ideas from [24, 48] and also develops novel technical ideas. A generalization of [24, 48] to time series data is to treat all observations $x_{it}$ independent and compute a sensitivity for each $x_{it}$ directly for importance sampling. However, this idea cannot capture the property that multiple observations $x_{i,1}, \ldots, x_{i,T_i}$ are drawn from the same Gaussian model (certain $l \in [k]$). To handle multiple observations, we show that although each $\psi_i$ consists of $T_i$ sub-functions $\psi_{it}$, it can be approximated by a single function on the average observation $b_i = \frac{\sum_{t \in [T_i]} x_{it}}{T_i}$, specifically, we have $\psi_i(\mu, I_d, 0_d) = d(b_i, \mu)^2 + O(1)$. This property enables us to "reduce" the representative complexity of $\psi_i$, and motivate two key steps of our construction: 1) For the sensitivity function, we give a reduction to a certain $k$-means clustering problem on average observations $b_i = \frac{\sum_{t \in [T_i]} x_{it}}{T_i}$ of entities (Definition 4.2) by upper bounding the maximum effect of covariances and autocorrelations and applying the fact that $\psi_i(\mu, I_d, 0_d) = d(b_i, \mu)^2 + O(1)$. 2) For the pseudo-dimension, we prove that there are only $\mathrm{poly}(k, d)$ intristic operators in $\psi_i$ between parameters $\theta$ and observations $x_{it}$, based on the reduction of the representative complexity of $\psi_i$.*

*Generalizing our results in the same manner as [26] does over [24, 48] would be an interesting future direction since we may get rid of Assumption 4.1. Currently, it is unclear how to generalize the approach of [26] to time series data since [26] assumes that each point is an integral point within a bounded box, while we consider a continuous GMM generative model (1), and hence, each coordinate of an arbitrary observation $x_{itr}$ is drawn from a certain continuous GMM distribution which is not integral with probability $\approx 1$ and can be unbounded.*

## 5 Empirical results

We compare the performance of our coreset algorithm CRGMM, with uniform sampling as a benchmark using synthetic data. The experiments are conducted with PyCharm IDE on a computer with 8-core CPU and 32 GB RAM.

**Datasets.** We generate two sets of **synthetic data** all with 250K observations with different number of entities $N$ and observations per individual $T_i$: (i) $N = T_i = 500$ for all $i \in [N]$ and (ii) $N = 200$, $T_i = 1250$ for all $i \in [N]$. As cross-entity variance is greater than within entity variance, this helps to assess coreset performance sensitivity when there are more entities (high $N$) vs. more frequent measurement/ longer measurement period (high $T_i$).

We fix $d = 2$, $k = 3$, $\lambda = 0.01$ and generate multiple datasets with model parameters randomly varied as follows: (i) Draw $\alpha$ from a uniform distribution over $\Delta_k$. (ii) For each $l \in [k]$, draw

$\mu^{(l)} \in \mathbb{R}^d$ from $N(0, I_d)$. (iii) Draw $\Sigma^{(l)}$ by first generating a random matrix $A \in [0, 1]^{d \times d}$ and then let $\Sigma^{(l)} = (AA^\top)^{-1}$, and (iv) draw all diagonal elements of $\Lambda^{(l)} \in \mathcal{D}^d$ from a uniform distribution over $[0, 1 - \sqrt{\lambda}]$. Given these draws of parameters, we generate a GMM time-series dataset as follows: For each $i \in [N]$, draw $l \in [k]$ given $\alpha$. Then, for all $t \in [T_i]$ draw $e_{it} \in \mathbb{R}^d$ with covariance matrix $\Sigma^{(l)}$ and autocorrelation matrix $\Lambda^{(l)}$ and compute $x_{it} = \mu^{(l)} + e_{it} \in \mathbb{R}^d$.

**Baseline and metrics.** We use coresets based on uniform sampling (**Uni**) and based on [48] (**LFKF**) that constructs a coreset for Gaussian mixture model with static data as the baseline. Given an integer $\Gamma$, uniformly sample a collection $S$ of $\Gamma$ entity-time pairs $(i, t) \in P_\mathcal{X}$, let $w(i) = \frac{N}{|I_S|}$ for each $i \in I_S$, and let $w^{(i)}(t) = \frac{T_i}{|J_{S,i}|}$ for each $t \in J_{S,i}$; and **LFKF** regard all entity-time pairs as independent points and sample a collection $S$ of $\Gamma$ entity-time pairs $(i, t) \in P_\mathcal{X}$ via importance sampling. For comparability, we set $\Gamma$ to be the same as our coreset size. The goal of selecting **LFKF** as a baseline is to see the effect of autocorrelations to the objective and show the difference between static clustering and time series clustering.

Let $V^\star$ (negative log-likelihood) denote the objective of Problem 3.1 with full data. Given a weighted subset $S \subseteq [N] \times [T]$, we first compute $(\alpha^\star, \theta^\star) := \arg\min_{\alpha \in \Delta_k, \theta \in \mathcal{P}_\lambda^k} f_S'(\alpha'(\theta), \theta) + \phi(\alpha, \theta)$ as the maximum likelihood solution over $S$.[2] Then $V_S^\star = f(\alpha^\star, \theta^\star)$ over the full dataset serves as a metric of model fit given model estimates from a weighted subset $S$. We use the likelihood ratio, i.e., $\gamma_S := 2(V_S^\star - V^\star)$ as a measure of the quality of $S$ [14]. The running time for GMM clustering with full data ($V^\star$) and coreset ($V_S^\star$) are $T_\mathcal{X}$ and $T_S$ respectively. $T_C$ is the coreset $S$ construction time.

**Empirical setup.** We vary $\varepsilon = 0.1, 0.2, 0.3, 0.4, 0.5$. For each $\varepsilon$, we run CRGMM and **Uni** to generate 5 coresets each. We estimate the model with the full dataset $\mathcal{X}$ and the coresets $S$ and record $V^\star, V_S^\star$ and the run times.

Table 1: Performance of $\varepsilon$-coresets for CRGMM w.r.t. varying $\varepsilon$. We report model fit as the mean of $V_S^\star$ (negative log-likelihood) w.r.t. 5 repetitions for our algorithm CRGMM and **Uni** on two synthetic datasets. "Synthetic 1" ($N = 500$ entities and $T_i = 500$ observations for each entity), and "Synthetic 2" ($N = 200$, $T_i = 1250$); Size ($\Gamma$): number of sampled entity-time pairs for CRGMM, **Uni**, and **LFKF**; $T_C$: construction time of coresets, which is at most 2s in our experiments. $T_S$ and $T_\mathcal{X}$: computation time for GMM clustering time over coresets and full data respectively.
Model fit $V_S^\star$ of our coreset is very close relative to the optimal $V^\star$ with full data; and much better than **Uni** and **LFKF**. Our coreset fit also does not decay as much as **Uni** and **LFKF** with larger error guarantees (0.1 to 0.5). Our coreset achieves a 3x-22x acceleration in solving Problem 3.1 compared to **Uni** and **LFKF** using same sample size and a 14x-171x acceleration relative to the full data.

| | $\varepsilon$ | $V_S^\star$ | | | $V^\star$ | $\gamma_S$ | | | size | $T_C + T_S$ (s) | | | $T_\mathcal{X}$ (s) |
|---|---|---|---|---|---|---|---|---|---|---|---|---|---|
| | | CRGMM | Uni | LFKF | | CRGMM | Uni | LFKF | | CRGMM | Uni | LFKF | |
| Synthetic 1 | 0.1 | 2050 | 2058 | 2069 | 2041 | 22 | 34 | 56 | 1514 | 109 | 2416 | 1355 | 3436 |
| | 0.2 | 2093 | 2210 | 2264 | | 104 | 342 | 446 | 404 | 41 | 1054 | 652 | |
| | 0.3 | 2194 | 2398 | 2384 | | 306 | 714 | 686 | 191 | 62 | 419 | 392 | |
| | 0.4 | 2335 | 3963 | 2705 | | 588 | 3844 | 1328 | 93 | 38 | 621 | 260 | |
| | 0.5 | 2383 | 3304 | 3461 | | 684 | 2526 | 2840 | 72 | 47 | 132 | 147 | |
| Synthetic 2 | 0.1 | 811 | 825 | 841 | 812 | 2 | 26 | 58 | 1718 | 694 | 1859 | 1687 | 9787 |
| | 0.2 | 824 | 895 | 871 | | 24 | 166 | 118 | 447 | 139 | 1991 | 1484 | |
| | 0.3 | 864 | 958 | 994 | | 104 | 292 | 364 | 199 | 51 | 527 | 832 | |
| | 0.4 | 860 | 1190 | 1114 | | 96 | 756 | 604 | 98 | 43 | 408 | 450 | |
| | 0.5 | 910 | 1361 | 1284 | | 196 | 1098 | 944 | 71 | 57 | 389 | 277 | |

**Results.** Table 1 summarizes the quality-size-time trade-offs of our coresets for different error guarantees $\varepsilon$. The $V_S^\star$ achieved by our coreset is always close to the optimal value $V^\star$ and is always smaller than that of **Uni** and **LFKF**. We next assess coreset size. For the $N = T_i = 500$ case, the coreset achieves a likelihood ratio $\gamma_S = 684$ with only 72 entity-time pairs (0.03%), but **Uni/LFKF** achieves the closest (but worse) likelihood ratio of $\gamma_S = 714/686$, with 191 entity-time pairs (0.08%). Thus, our coreset achieves a better likelihood than uniform and **LFKF** with less than 40% of observations. Further, from Figure 1, we see that not only is the quality of our coreset ($\gamma_S$) superior, it has lower standard deviations suggesting lower variance in performance. In terms of total computation time relative to the full data ($\frac{T_\mathcal{X}}{T_C + T_S}$), our coresets speed up by 14x-171x. Also, the

---

[2] We solve this optimization problem using an EM algorithm similar to [5]. The M step in each iteration is based on IRLS [40] and the E-step involves an individual level Bayesian update for $\alpha$.

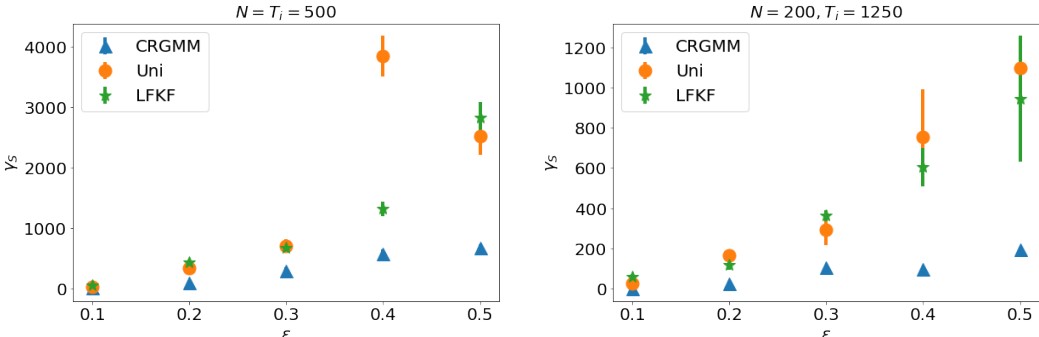

Figure 1: Mean +/- std. of $\gamma_S$, w.r.t. varying $\varepsilon$. $\gamma_S$ is the likelihood ratio (LR) for estimates from coresets relative to full data. CRGMM not only has better fit than **Uni** and **LFKF**, but much small std. in performance. The std. for our coreset is even smaller for Synthetic 2 with fewer entities.

computation time $(T_C + T_S)$ of **Uni** and **LFKF** is always larger than that of our coreset (3x-22x).[3] Finally, $V_S^\star$ is lower for Synthetic 2 given that there are fewer entities and cross-entity variance is greater than within entity variance. From Figure 1, we also see that both the fit and std in performance for coresets is much lower for Synthetic 2. Thus, overall our coreset performs better when there is more time series data relative to entities ($T_i \gg N$)—a feature of sensor based measurement that motivates this paper.

# 6 Limitations, conclusion, and future work

In this paper, we study the problem of constructing coresets for clustering problems with time series data; in particular, we address the problem of constructing coresets for time series data generated from Gaussian mixture models with auto-correlations across time. Our coreset construction algorithm is efficient under a mild boundedness assumption on the covariance matrices of the underlying Gaussians, and the size of the coreset is independent of the number of entities and the number of observations and depends only polynomially on the number of clusters, the number of variables and an error parameter. Through empirical analysis on synthetic data, we demonstrate that the coreset sampling is superior to uniform sampling in computation time and accuracy.

Our work leaves several interesting directions for future work on time series clustering. While our current specification with autocorrelations assumes a stable time series pattern over time, future work should extend it to a hidden Markov process, where the time series process may switch over time. Further, while our focus here is on model-based clustering, it would be useful to consider how coresets should be constructed for other clustering methods such as direct clustering of raw data and indirect clustering of features.

Overall, we hope the paper stimulates more research on coreset construction for time series data on a variety of unsupervised and supervised machine learning algorithms. The savings in storage and computational cost without sacrificing accuracy is not only financially valuable, but also can have sustainability benefits through reduced energy consumption. Finally, recent research has shown that summaries (such as coresets) for static data need to include fairness constraints to avoid biased outcomes for under-represented groups based on gender and race when using the summary [16, 36]; future research needs to extend such techniques for time series coresets.

## Acknowledgments

This research was supported in part by an NSF CCF-1908347 grant.

---

[3]A possible explanation is that our coreset selects $I_S$ of entities that are more representative that **Uni**, and hence, achieves a better convergence speed.

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
