# A Proof of Lemma 4.4: The first stage of Algorithm 1 outputs an entity-level coreset

For preparation, we first introduce an importance sampling framework for coreset construction, called the Feldman-Langberg framework [27, 13].

## A.1 The Feldman-Langberg framework

We first give the definition of query space and the corresponding coresets.

**Definition A.1** (**Query space [27, 13]**)**.** *Let $\mathcal{X}$ be a finite set together with a weight function $u : \mathcal{X} \rightarrow \mathbb{R}_{\geq 0}$. Let $\mathcal{P}$ be a set called queries, and $f_x : \mathcal{P} \rightarrow \mathbb{R}_{\geq 0}$ be a given loss function w.r.t. $x \in \mathcal{X}$. The total cost of $\mathcal{X}$ with respect to a query $\theta \in \mathcal{P}$ is $f(\theta) := \sum_{x \in \mathcal{X}} u(x) \cdot f_x(\theta)$. The tuple $(\mathcal{X}, u, \mathcal{P}, f)$ is called a query space. Specifically, if $u(x) = 1$ for all $x \in \mathcal{X}$, we use $(\mathcal{X}, \mathcal{P}, f)$ for simplicity.*

Intuitively, $f$ represents a linear combination of weighted functions indexed by $\mathcal{X}$, and $\mathcal{P}$ represents the ground set of $f$. Due to the separability of $f$, we have the following coreset definition.

**Definition A.2** (**Coresets of a query space [27, 13]**)**.** *Let $(\mathcal{X}, u, \mathcal{P}, f)$ be a query space and $\varepsilon \in (0, 1)$ be an error parameter. An $\varepsilon$-coreset of $(\mathcal{X}, u, \mathcal{P}, f)$ is a weighted set $S \subseteq \mathcal{X}$ together with a weight function $w : S \rightarrow \mathbb{R}_{\geq 0}$ such that for any $\theta \in \mathcal{P}$, $\sum_{x \in S} w(x) \cdot f_x(\theta) \in (1 \pm \varepsilon) \cdot f(\theta)$.*

**Remark A.3.** *For instance, we set $\mathcal{X} = [N]$, $u = 1$, $\mathcal{P} = \Delta_k \times \mathcal{P}_\lambda^k$ and $f = f'$ in the GMM time-series clustering problem. Then by Definition A.2, Lemma 4.4 represents that $(I_S, w)$ is an $\varepsilon$-coreset for the query space $([N], 1, \Delta_k \times \mathcal{P}_\lambda^k, f')$.*

*Another example is to set $\mathcal{X} = [T_i]$, $u = 1$, $\mathcal{P} = \mathcal{P}_\lambda$ and $f = \psi_i$. Then Lemma 4.5 represents that $(J_{S,i}, w^{(i)})$ is an $\varepsilon$-coreset for the query space $([T_i], 1, \mathcal{P}_\lambda, \psi_i)$.*

Now we are ready to give the Feldman-Langberg framework.

**The Feldman-Langberg framework.** Feldman and Langberg [27] show how to construct coresets by importance sampling and the coreset size has been improved by [13]. For preparation, we first give the notion of sensitivity which measures the maximum influence for each point $x \in \mathcal{X}$.

**Definition A.4** (**Sensitivity [27, 13]**)**.** *Given a query space $(\mathcal{X}, u, \mathcal{P}, f)$, the sensitivity of a point $x \in \mathcal{X}$ is $s(x) := \sup_{\theta \in \mathcal{P}} \frac{u(x) \cdot f_x(\theta)}{f(\mathcal{X}, u, \theta)}$. The total sensitivity of the query space is $\sum_{x \in \mathcal{X}} s(x)$.*

We also introduce a notion which measures the combinatorial complexity of a query space.

**Definition A.5** (**Pseudo-dimension [27, 13]**)**.** *For a query space $(\mathcal{X}, u, \mathcal{P}, f)$, we define $\mathsf{range}(\theta, r) = \{x \in \mathcal{X} : u(x) \cdot f_x(\theta) \leq r\}$ for every $\theta \in \mathcal{P}$ and $r \geq 0$. The (pseudo-)dimension of $(\mathcal{X}, u, \mathcal{P}, f)$ is the largest integer $t$ such that there exists a subset $A \subseteq \mathcal{X}$ of size $t$ satisfying that $|\{A \cap \mathsf{range}(\theta, r) : \theta \in \mathcal{P}, r \geq 0\}| = 2^{|A|}$.*

Pseudo-dimension plays the same role as VC-dimension [56]. Specifically, if the range of $f$ is $\{0, 1\}$ and $u = 1$, pseudo-dimension can be regarded as a generalization of VC-dimension to function spaces. Now we are ready to describe the Feldman-Langberg framework.

**Theorem A.6** (**Feldman-Langberg framework [27, 13]**)**.** *Let $(\mathcal{X}, u, \mathcal{P}, f)$ be a given query space and $\varepsilon, \delta \in (0, 1)$. Let $\dim$ be an upper bound of the pseudo-dimension of every query space $(\mathcal{X}, u', \mathcal{P}, f)$ over $u'$. Suppose $s : \mathcal{X} \rightarrow \mathbb{R}_{\geq 0}$ is a function satisfying that for any $x \in \mathcal{X}$, $s(x) \geq \sup_{\theta \in \mathcal{P}} \frac{u(x) \cdot f_x(\theta)}{f(\mathcal{X}, u, \theta)}$, and define $\mathcal{G} := \sum_{x \in \mathcal{X}} s(x)$ to be the total sensitivity. Let $S \subseteq \mathcal{X}$ be constructed by taking $O\left(\varepsilon^{-2} \mathcal{G}(\dim \cdot \ln \mathcal{G} + \ln(1/\delta))\right)$ samples, where each sample $x \in \mathcal{X}$ is selected with probability $\frac{s(x)}{\mathcal{G}}$ and has weight $w(x) := \frac{\mathcal{G}}{|S| \cdot s(x)}$. Then, with probability at least $1 - \delta$, $S$ is an $\varepsilon$-coreset of $(\mathcal{X}, u, \mathcal{P}, f)$.*

## A.2 Bounding the pseudo-dimension of $f'$

Our proof idea is similar to that in [48]. For preparation, we need the following lemma which is proposed to bound the pseudo-dimension of feed-forward neural networks.

**Lemma A.7 (Restatement of Theorem 8.14 of [4]).** *Let $(\mathcal{X}, u, \mathcal{P}, f)$ be a given query space where $f_x(\theta) \in \{0, 1\}$ for any $x \in \mathcal{X}$ and $\theta \in \mathcal{P}$, and $\mathcal{P} \subseteq \mathbb{R}^m$. Suppose that $f$ can be computed by an algorithm that takes as input the pair $(x, \theta) \in \mathcal{X} \times \mathcal{P}$ and returns $f_x(\theta)$ after no more than $l$ of the following operations:*

- *the exponent function $a \to e^a$ on real numbers.*

- *the arithmetic operations $+, -, \times,$ and $/$ on real numbers.*

- *jumps conditioned on $>, \geq, <, \leq, =,$ and $\neq$ comparisons of real numbers, and*

- *output 0,1.*

*If the $l$ operations include no more than $q$ in which the exponential function is evaluated, then the pseudo-dimension of $(\mathcal{X}, u, \mathcal{P}, f)$ is at most $O(m^2q^2 + mq(l + \ln mq))$.*

Note that the above lemma requires that the range of functions $f_x$ is $[0, 1]$. We have the following lemma which can help extend this range to $\mathbb{R}$.

**Lemma A.8 (Restatement of Lemma 4.1 of [58]).** *Let $(\mathcal{X}, u, \mathcal{P}, f)$ be a given query space. Let $g_x : \mathcal{P} \times \mathbb{R} \to \{0, 1\}$ be the indicator function satisfying that for any $x \in \mathcal{X}$, $\theta \in \mathcal{P}$ and $r \in \mathbb{R}$,*

$$g_x(\theta, r) = I\left[u(x) \cdot f(x, \theta) \geq r\right].$$

*Then the pseudo-dimension of $(\mathcal{X}, u, \mathcal{P}, f)$ is precisely the pseudo-dimension of the query space $(\mathcal{X}, u, \mathcal{P} \times \mathbb{R}, g_f)$.*

Now we are ready to prove bound the pseudo-dimension of $f'$ by the following lemma.

**Lemma A.9 (Pseudo-dimension of $f'$).** *The pseudo-dimension of $([N], u, \Delta_k \times \mathcal{P}_\lambda^k, f')$ over weight functions $u : [N] \to \mathbb{R}_{\geq 0}$ is at most $O(k^4d^4 + k^3d^8)$.*

*Proof.* Our argument is similar to that in [37, Lemma 5.9]. Fix a weight function $u : [N] \to \mathbb{R}_{\geq 0}$. We only need to consider the following indicator function $g_i : \Delta_k \times \mathcal{P}_\lambda^k \times \mathbb{R}_{\geq 0} \to \{0, 1\}$ where for any $\alpha \in \Delta_k$, $\theta \in \mathcal{P}_\lambda^k$ and $r \in \mathbb{R}_{\geq 0}$,

$$g_i(\alpha, \theta, r) := I\left[\sum_{l \in [k]} \alpha_l \cdot \exp(-\frac{1}{2T_i}\psi_i(\mu^{(l)}, \Sigma^{(l)}, \Lambda^{(l)})) \geq r\right].$$

Note that the parameter space is $\Delta_k \times \mathcal{P}_\lambda^k$ which consists of at most $m = O(kd^2)$ parameters. For any $(\mu, \Sigma, \Lambda) \in \mathcal{P}_\lambda$, function $\psi_i(\mu, \Sigma, \Lambda)$ can be represented as a multivariate polynomial that consists of $O(d^6)$ terms $\mu_{c_1}^{b_1}\mu_{c_2}^{b_2}\Lambda_{c_3,c_3}^{b_3}\Lambda_{c_4,c_4}^{b_4}\left(\Sigma^{-1}\right)_{c_5,c_6}^{b_5}$ where $c_1, c_2, c_3, c_4, c_5, c_6 \in [d]$, and $b_1, b_2, b_3, b_4, b_5 \in \{0, 1\}$. Thus, $g_i$ consists of $l = O(kd^6)$ arithmetic operations, $q = k$ exponential functions, and $k$ jumps. By Lemmas A.7 and A.8, we complete the proof. $\qquad\square$

### A.3 Bounding the total sensitivity of $f'$

Next, we prove that function $s$ (Line 6 of Algorithm 1) is a sensitivity function w.r.t. $f'$; summarized as follows.

**Lemma A.10 ($s$ is a sensitivity function w.r.t. $f'$).** *For each $i \in [N]$, we have*

$$s(i) \geq \max_{\alpha \in \Delta_k, \theta \in \mathcal{P}_\lambda^k} \frac{f_i'(\alpha, \theta)}{f'(\alpha, \theta)}.$$

*Moreover, $\sum_{i \in [N]} s(i) \leq (16D + 12Dk)/\lambda$.*

To prove the lemma, we will use a reduction from general $\Sigma$ to $I_d$ and from $\Lambda$ to $0_d$ (without both covariances and autocorrelations), which upper bounds the affect of the covariance matrix and autocorrelation matrices. We define $\psi_i^{(O)} : \mathbb{R}^d \to \mathbb{R}_{\geq 0}$ to be

$$\psi_i^{(O)}(\mu) := \sum_{t \in [T_i]} \|x_{it} - \mu\|_2^2 \tag{3}$$

for any $\mu \in \mathbb{R}^d$, and define $f^{(O)} : \Delta_k \times \mathbb{R}^{d \times k} \to \mathbb{R}_{\geq 0}$ to be

$$
\begin{aligned}
f^{(O)}(\alpha, \theta^{(O)}) &:= \sum_{i \in [N]} f_i^{(O)}(\alpha, \theta^{(O)}) \\
&= -\sum_{i \in [N]} \ln \sum_{l \in [k]} \alpha_l \cdot \exp\left( -\frac{1}{2T_i \cdot \min_{l' \in [k]} \lambda_{\min}(\Sigma^{(l')})} \psi_i^{(O)}(\mu^{(l)}) \right)
\end{aligned}
$$

for any $\alpha \in \Delta_k$ and $\theta^{(O)} = (\mu^{(l)})_{l \in [k]} \in \mathbb{R}^{d \times k}$. Compared to $f'$, we note that $f^{(O)}$ does not contain covariance and autocorrelation matrices.

**Clustering cost of entities.** By the definition of $\psi_i^{(O)}$, we have that for any $\mu \in \mathbb{R}^d$,

$$
\frac{1}{T_i} \psi_i^{(O)}(\mu) = \|\frac{\sum_{t \in [T_i]} x_{it}}{T_i} - \mu\|_2^2 + \frac{1}{T_i} \sum_{t \in [T_i]} \|x_{it}\|_2^2 - \frac{\|\sum_{t \in [T_i]} x_{it}\|_2^2}{T_i^2}. \tag{4}
$$

Next, we introduce another function $s^{(O)} : [N] \to \mathbb{R}_{\geq 0}$ as a sensitivity function w.r.t. $f^{(O)}$, i.e., for any $i \in [N]$,

$$
s^{(O)}(i) := \max_{\alpha \in \Delta_k, \theta^{(O)} \in \mathbb{R}^{d \times k}} \frac{f_i^{(O)}(\alpha, \theta^{(O)})}{f^{(O)}(\alpha, \theta^{(O)})}.
$$

Define $\mathcal{G}^{(O)} := \sum_{i \in [N]} s^{(O)}(i)$ to be the total sensitivity w.r.t. $f^{(O)}$. We first have the following lemma.

**Lemma A.11** (**Relation between sensitivities w.r.t. $f_i^{(O)}$ and $f_i'$**)**.** *For each $i \in [N]$, we have*

$$
s^{(O)}(i) \leq \max_{\alpha \in \Delta_k, \theta \in \mathcal{P}_\lambda^k} \frac{f_i'(\alpha, \theta)}{f'(\alpha, \theta)} \leq 4D \cdot s^{(O)}(i)/\lambda.
$$

*Proof.* It is easy to verify $s^{(O)}(i) \leq \max_{\alpha \in \Delta_k, \theta \in \mathcal{P}_\lambda^k} \frac{f_i'(\alpha, \theta)}{f'(\alpha, \theta)}$ since

$$
\begin{aligned}
&\max_{\alpha \in \Delta_k, \theta \in \mathcal{P}_\lambda^k} \frac{f_i'(\alpha, \theta)}{f'(\alpha, \theta)} \\
&\geq \max_{\alpha \in \Delta_k, \theta \in (\mathbb{R}^d \times I_d \times 0_d)^k} \frac{f_i'(\alpha, \theta)}{f'(\alpha, \theta)} & (0_d \in \mathcal{D}_\lambda^d, I_d \in \mathcal{S}^d) \\
&= \max_{\alpha \in \Delta_k, \theta^{(O)} \in \mathbb{R}^{d \times k}} \frac{f_i^{(O)}(\alpha, \theta^{(O)})}{f^{(O)}(\alpha, \theta^{(O)})} & (\text{Defn. of } f_i^{(O)}) \\
&= s^{(O)}(i).
\end{aligned}
$$

For the other side, we have the following claim that for any $i \in [N]$ and $\theta = (\mu, \Sigma, \Lambda) \in \mathcal{P}_\lambda$,

$$
\frac{\lambda}{\lambda_{\max}(\Sigma)} \cdot \psi_i^{(O)}(\mu) \leq \psi_i(\mu, \Sigma, \Lambda) \leq \frac{4}{\lambda_{\min}(\Sigma)} \cdot \psi_i^{(O)}(\mu). \tag{5}
$$

Then for any $\alpha \in \Delta_k$ and $\theta = (\mu^{(l)}, \Sigma^{(l)}, \Lambda^{(l)})_{l \in [k]} \in \mathcal{P}_\lambda^k$, letting $\theta^{(O)} = (\mu^{(l)})_{l \in [k]} \in \mathbb{R}^{d \times k}$ and $\beta = \min_{l' \in [k]} \lambda_{\min}(\Sigma^{(l')})^2$, we have

$$
\begin{aligned}
\frac{f_i'(\alpha, \theta)}{f'(\alpha, \theta)} &= \frac{-\ln \sum_{l \in [k]} \alpha_l \cdot \exp(-\frac{1}{2T_i} \psi_i(\mu^{(l)}, \Sigma^{(l)}, \Lambda^{(l)}))}{-\sum_{j \in [N]} \ln \sum_{l \in [k]} \alpha_l \cdot \exp(-\frac{1}{2T_i} \psi_j(\mu^{(l)}, \Sigma^{(l)}, \Lambda^{(l)}))} && \text{(by definition)} \\[2mm]
&\leq \frac{-\ln \sum_{l \in [k]} \alpha_l \cdot \exp(-\frac{2}{T_i \lambda_{\min}(\Sigma^{(l)})} \psi_i^{(O)}(\mu^{(l)}))}{-\sum_{j \in [N]} \ln \sum_{l \in [k]} \alpha_l \cdot \exp(-\frac{\lambda}{2T_i \lambda_{\max}(\Sigma^{(l)})} \cdot \psi_j^{(O)}(\mu^{(l)}))} && \text{(Ineq. (5))} \\[2mm]
&\leq \frac{4 f_i^{(O)}(\alpha, \theta^{(O)})}{-\sum_{j \in [N]} \ln \left( \sum_{l \in [k]} \alpha_l \cdot \exp(-\frac{1}{2T_i \cdot \beta} \cdot \psi_j^{(O)}(\mu^{(l)})) \right)^{\frac{\lambda\beta}{\lambda_{\max}(\Sigma^{(l)})}}} && \text{(Defn. of } f_i^{(O)}) \\[2mm]
&\leq \frac{4 f_i^{(O)}(\alpha, \theta^{(O)})}{-\sum_{j \in [N]} \ln \left( \sum_{l \in [k]} \alpha_l \cdot \exp(-\frac{1}{2T_i \cdot \beta} \cdot \psi_j^{(O)}(\mu^{(l)})) \right)^{\lambda/D}} && \text{(Assumption 4.1)} \\[2mm]
&= \frac{4D \cdot f_i^{(O)}(\alpha, \theta^{(O)})}{\lambda \cdot f^{(O)}(\alpha, \theta^{(O)})}. && \text{(by definition)}
\end{aligned}
$$

Consequently, we have $\max_{\alpha \in \Delta_k, \theta \in \mathcal{P}_\lambda^k} \frac{f_i'(\alpha, \theta)}{f'(\alpha, \theta)} \leq 4D \cdot s^{(O)}(i)/\lambda$, which completes the proof.

**Proof of Claim** (5). It remains to prove Claim (5). For any $i \in [N]$ and $\theta = (\mu, \Sigma, \Lambda) \in \mathcal{P}_\lambda$, we have

$$
\begin{aligned}
&\psi_i(\mu, \Sigma, \Lambda) \\
={}& (x_{i,1} - \mu)^\top \Sigma^{-1} (x_{i,1} - \mu) - (\Lambda(x_{i,1} - \mu))^\top \Sigma^{-1} (\Lambda(x_{i,1} - \mu)) \\
&+ \sum_{t=2}^{T_i} ((x_{it} - \mu) - \Lambda(x_{i,t-1} - \mu))^\top \Sigma^{-1} ((x_{it} - \mu) - \Lambda(x_{i,t-1} - \mu)) \\
\in{}& [\frac{1}{\lambda_{\max}(\Sigma)}, \frac{1}{\lambda_{\min}(\Sigma)}] \cdot \left( (x_{i,1} - \mu)^\top (x_{i,1} - \mu) - (\Lambda(x_{i,1} - \mu))^\top (\Lambda(x_{i,1} - \mu)) \right) \\
&+ [\frac{1}{\lambda_{\max}(\Sigma)}, \frac{1}{\lambda_{\min}(\Sigma)}] \cdot \sum_{t=2}^{T_i} ((x_{it} - \mu) - \Lambda(x_{i,t-1} - \mu))^\top ((x_{it} - \mu) - \Lambda(x_{i,t-1} - \mu)).
\end{aligned}
$$

Hence, it suffices to prove that

$$
\begin{aligned}
&(x_{i,1} - \mu)^\top \Sigma^{-1} (x_{i,1} - \mu) - (\Lambda(x_{i,1} - \mu))^\top \Sigma^{-1} (\Lambda(x_{i,1} - \mu)) \\
&+ \sum_{t=2}^{T_i} ((x_{it} - \mu) - \Lambda(x_{i,t-1} - \mu))^\top ((x_{it} - \mu) - \Lambda(x_{i,t-1} - \mu)) \\
\in{}& [\lambda, 4] \cdot \psi_i^{(O)}(\mu)
\end{aligned}
$$

Since $\Lambda \in \mathcal{D}_\lambda^d$, we suppose $\Lambda = (\Delta_1, \ldots, \Delta_d)$. Then we have

$$
\begin{aligned}
&(x_{i,1} - \mu)^\top (x_{i,1} - \mu) - (\Lambda(x_{i,1} - \mu))^\top (\Lambda(x_{i,1} - \mu)) \\
&+ \sum_{t=2}^{T_i} ((x_{it} - \mu) - \Lambda(x_{i,t-1} - \mu))^\top ((x_{it} - \mu) - \Lambda(x_{i,t-1} - \mu)) \\
={}& \sum_{r \in [d]} (1 - \Delta_r^2)(x_{i1r} - \mu_r)^2 \\
&+ \sum_{t=2}^{T_i} ((x_{itr} - \mu_r) - \Delta_r(x_{i,t-1,r} - \mu_r))^\top ((x_{itr} - \mu_r) - \Delta_r(x_{i,t-1,r} - \mu_r)).
\end{aligned}
$$

On one hand, we have

$$(1 - \Delta_r^2)(x_{i1r} - \mu_r)^2 + \sum_{t=2}^{T_i} \left((x_{itr} - \mu_r) - \Delta_r(x_{i,t-1,r} - \mu_r)\right)^2$$

$$\leq \quad (1 - \Delta_r^2)(x_{i1r} - \mu_r)^2 + \sum_{t=2}^{T_i} 2\left((x_{itr} - \mu_r)^2 + \Delta_r^2(x_{i,t-1,r} - \mu_r)^2\right) \quad \text{(Arithmetic Ineq.)}$$

$$\leq \quad 4 \sum_{t \in [T_i]} (x_{itr} - \mu_r)^2. \qquad\qquad (\Delta_r \leq 1)$$

On the other hand, we have

$$(1 - \Delta_r^2)(x_{i1r} - \mu_r)^2 + \sum_{t=2}^{T_i} \left((x_{itr} - \mu_r) - \Delta_r(x_{i,t-1,r} - \mu_r)\right)^2$$

$$= \quad (x_{i1r} - \mu_r)^2 + \sum_{t=2}^{T_i}(1 + \Delta_r^2)(x_{itr} - \mu_r)^2 - 2\Delta_r(x_{itr} - \mu_r)(x_{i,t-1,r} - \mu_r)$$

$$\geq \quad (1 - \Delta_r)^2 \sum_{t \in [T_i]} (x_{itr} - \mu_r)^2 \quad \text{(Arithmetic Ineq.)}$$

$$\geq \quad \lambda \sum_{t \in [T_i]} (x_{itr} - \mu_r)^2. \quad \text{(Assumption 4.1)}$$

This completes the proof. $\qquad\qquad\qquad\qquad\qquad\qquad\qquad\qquad\qquad\qquad \square$

By the definition of $s$ and the above lemma, it suffices to prove the following lemma that provides an upper bound for $s^{(O)}$.

**Lemma A.12 (Sensitivities w.r.t. $f^{(O)}$).** *The following holds:*

1. *For each $i \in [N]$, we have*

$$s^{(O)}(i) \leq \frac{4\|b_i - c_{p(i)}^\star\|_2^2}{\mathsf{OPT}^{(O)} + A} + 3s^c(i)$$

2. $\mathcal{G}^{(O)} \leq 4 + 3k.$

For preparation, we introduce some notations related to the clustering problem (Definition 4.2). For any $\mu \in \mathbb{R}^d$, we define

$$h_i^c(\mu) := \|c_{p(i)}^\star - \mu\|_2^2,$$

and for any $\alpha \in \Delta_k, \theta^{(O)} \in \mathbb{R}^{d \times k}$,

$$f_i^c(\alpha, \theta^{(O)}) := -\ln \sum_{l \in [k]} \alpha_k \cdot \exp\left(-\frac{1}{2T_i \cdot \beta} h_i^c(\mu^{(l)})\right),$$

where $\beta = \min_{l' \in [k]} \lambda_{\min}(\Sigma^{(l')})^2$. Let $f^c := \sum_{i \in [N]} f_i^c$. Then similarly, we can prove that $s^c$ (Line 5 of Algorithm 1) is a sensitivity function w.r.t. $f^c$; summarized as follows.

**Lemma A.13 ($s^c$ is a sensitivity function w.r.t. $f^c$).** *For each $i \in [N]$,*

$$s^c(i) \geq \max_{\alpha \in \Delta_k, \theta^{(O)}} \frac{f_i^c(\alpha, \theta^{(O)})}{f^c(\alpha, \theta^{(O)})}.$$

*Moreover, $\sum_{i \in [N]} s^c(i) \leq k$.*

*Proof.* This lemma is a direct corollary by the fact that there are only $k$ different centers $C_i^\star$, which implies that there are at most $k$ different functions $f_i^c$ accordingly. We partition $[N]$ into at most

$k$ groups $A_l$ where each element $i \in A_l$ satisfies that $c_{p(i)}^\star = l$. Then we observe that $f_i^c = f_j^c$ if $i, j \in A_l$. Then for any $\alpha, \theta^{(O)} \in \Delta_k \times \mathbb{R}^{d \times k}$,

$$\frac{f_i^c(\alpha, \theta^{(O)})}{f^c(\alpha, \theta^{(O)})} \leq \frac{f_i^c(\alpha, \theta^{(O)})}{\sum_{j \in A_l} f_j^c(\alpha, \theta^{(O)})} = \frac{1}{|A_l|} \leq s^c(i),$$

which implies the lemma. $\qquad\square$

To prove Lemma A.12, the main idea is to relate $s^{(O)}(i)$ to $s^c(i)$. The idea is similar to [48]. For preparation, we also need the following key observation.

**Lemma A.14** (**Upper bounding the projection cost**). *For a fixed number $L > 0$ and a fixed $\theta = (\alpha, \theta^{(O)}) \in \Delta_k \times \mathbb{R}^{d \times k}$ and a fixed value $a \geq 0$, define $\pi_{a,\theta} : \mathbb{R}^d \to \mathbb{R}_{\geq 0}$ as*

$$\pi_{a,\theta,L}(y) = -\ln \sum_{l \in [k]} \alpha_l \cdot \exp\left(-\frac{1}{2L}\left(\|y - \mu^{(l)}\|_2^2 + a\right)\right).$$

*Then, for every $y, y' \in \mathbb{R}^d$ it holds that*

$$\pi_{a,\theta}(y) \leq \frac{2}{L}\|y - y'\|_2^2 + \pi_{a,\theta,L}(y').$$

*Proof.* Use the relaxed triangle inequality for $l_2^2$-norm, we have

$$
\begin{aligned}
\pi_{a,\theta,L}(y) = \quad & -\ln \sum_{l \in [k]} \alpha_l \cdot \exp\left(-\frac{1}{2L}\left(\|y - \mu^{(l)}\|_2^2 + a\right)\right) \\
\leq \quad & -\ln \sum_{l \in [k]} \alpha_l \cdot \exp\left(-\frac{2}{L}\|y' - \mu^{(l)}\|_2^2 + \frac{1}{2L}\left(\|y - y'\|_2^2 + a\right)\right) \\
& \text{(relaxed triangle ineq.)} \\
\leq \quad & -\ln\left(\exp\left(-\frac{2}{L}\|y - y'\|_2^2\right) \cdot \sum_{l \in [k]} \alpha_l \cdot \exp\left(-\frac{1}{2L}\left(\|y' - \mu^{(l)}\|_2^2 + a\right)\right)\right) \\
\leq \quad & \frac{2}{L}\|y - y'\|_2^2 - \ln \sum_{l \in [k]} \alpha_l \cdot \exp\left(-\frac{1}{2L}\left(\|y' - \mu^{(l)}\|_2^2 + a\right)\right) \\
\leq \quad & \frac{2}{L}\|y - y'\|_2^2 - \ln\left(\sum_{l \in [k]} \alpha_l \cdot \exp\left(-\frac{1}{2L}\left(\|y' - \mu^{(l)}\|_2^2 + a\right)\right)\right)^2 \\
= \quad & \frac{2}{L}\|y - y'\|_2^2 + \pi_{a,\theta,L}(y').
\end{aligned}
$$

$\qquad\square$

Recall that $b_i \leftarrow \frac{\sum_{t \in [T_i]} x_{it}}{T_i}$. Now we are ready to prove Lemma A.12.

*Proof of Lemma A.12.* For each $i \in [N]$ and $\theta = (\alpha, \theta^{(O)}) \in \Delta_k \times \mathbb{R}^{d \times k}$, letting $\beta = \min_{l' \in [k]} \lambda_{\min}(\Sigma^{(l')})^2$, we have

$$
\begin{aligned}
& f_i^{(O)}(\theta) \\
=\ & \pi_{a_i,\theta,\beta}(b_i) \\
\leq\ & \frac{2}{\beta}\|b_i - c_{p(i)}^\star\|_2^2 + \pi_{a_i,\theta,\beta}(c_{p(i)}^\star) && \text{(Lemma A.14)} \\
\leq\ & \frac{2}{\beta}\|b_i - c_{p(i)}^\star\|_2^2 + s^c(i) \cdot \sum_{j \in [N]} \pi_{a_i,\theta,\beta}(c_{p(j)}^\star) && \text{(Defns. of } s^c\text{)} \\
\leq\ & \frac{2}{\beta}\|b_i - c_{p(i)}^\star\|_2^2 + s^c(i) \cdot \sum_{j \in [N]} \left( \frac{2}{\beta} \cdot \|b_j - c_{p(j)}^\star\|_2^2 + \pi_{a_i,\theta,\beta}(b_j) \right) && \text{(Lemma A.14)} \\
\leq\ & \frac{2}{\beta}\|b_i - c_{p(i)}^\star\|_2^2 + s^c(i) \cdot \left( \frac{2 \cdot \mathsf{OPT}^{(O)}}{\beta} + f^{(O)}(\theta) \right). && \text{(Defns. of } \mathsf{OPT}^{(O)}\text{)}
\end{aligned}
$$

(6)

Let $\mathsf{OPT} := \min_\theta \sum_{i \in [N]} f_i^{(O)}(\theta)$. We have that

$$
\begin{aligned}
& \mathsf{OPT} \\
=\ & \min_\theta \sum_{i \in [N]} f_i^{(O)}(\theta) \\
=\ & \min_\theta \sum_{i \in [N]} -\ln \sum_{l \in [k]} \alpha_l \cdot \exp\left( -\frac{1}{2\beta}\left( \|b_i - \mu^{(l)}\|_2^2 + a_i \right) \right) \\
\geq\ & \min_\theta \sum_{i \in [N]} -\ln \sum_{l \in [k]} \alpha_l \cdot \exp\left( -\frac{1}{2\beta}\left( \min_{l' \in [k]} \|b_i - \mu^{(l')}\|_2^2 + a_i \right) \right) \\
\geq\ & \min_\theta \sum_{i \in [N]} -\ln \sum_{l \in [k]} \alpha_l \cdot \exp\left( -\frac{1}{2\beta}\left( \min_{l' \in [k]} \|b_i - \mu^{(l')}\|_2^2 + a_i \right) \right) \\
=\ & \frac{1}{2\beta}\left( \min_\theta \sum_{i \in [N]} \min_{l' \in [k]} \|b_i - \mu^{(l')}\|_2^2 + a_i \right) \\
\geq\ & \frac{1}{2\beta} \cdot \left( \mathsf{OPT}^{(O)} + A \right). && \text{(Defns. of } \mathsf{OPT}^{(O)} \text{ and } A\text{)}
\end{aligned}
$$

(7)

Hence, we have

$$
\begin{aligned}
s^{(O)}(i) =\ & \max_{\theta \in \Delta_k \times \mathbb{R}^{d \times k}} \frac{f_i^{(O)}(\theta)}{f^{(O)}(\theta)} \\
\leq\ & \frac{2 \cdot \|b_i - c_{p(i)}^\star\|_2^2}{\beta \cdot \mathsf{OPT}} + \frac{s^c(i) \cdot \mathsf{OPT}^{(O)}}{\beta \cdot \mathsf{OPT}} + s^c(i) && \text{(Ineq. (6))} \\
\leq\ & \frac{4 \cdot \|b_i - c_{p(i)}^\star\|_2^2}{\mathsf{OPT}^{(O)} + A} + 3s^c(i). && \text{(Ineq. (7))}
\end{aligned}
$$

The second property is a direct conclusion. $\qquad\square$

Now we are ready to prove Lemma A.10.

*Proof of Lemma A.10.* Lemma A.10 is a direct corollary of Lemmas A.11 and A.12 since for each $i \in [N]$,

$$
\begin{aligned}
\max_{\alpha \in \Delta_k, \theta \in \mathcal{P}_\lambda^k} \frac{f_i'(\alpha, \theta)}{f'(\alpha, \theta)} \leq\ & s^{(O)}(i)/\lambda && \text{(Lemma A.11)} \\
\leq\ & 4D \left( \frac{4 \cdot \|b_i - c_{p(i)}^\star\|_2^2}{\mathsf{OPT}^{(O)} + A} + 3s^c(i) \right)/\lambda && \text{(Lemma A.12)} \\
=\ & s(i). && \text{(Line 6 of Algorithm 1)}
\end{aligned}
$$

$\square$

By the Feldman-Langberg framework (Theorem A.6), we note that Lemma 4.4 is a direct corollary of Lemmas A.9 and A.10.

# B   Proof of Lemma 4.5: The second stage of Algorithm 1 outputs a time-level coreset

The proof idea is similar to that in Lemma 4.4, i.e., to bound the pseudo-dimension and the total sensitivity for the query space $([T_i], 1, \mathcal{P}_\lambda, \psi_i)$.

## B.1   Bounding the pseudo-dimension of $\psi_i$

We have the following lemma.

**Lemma B.1** (**Pseudo-dimension of $\psi_i$**). *The pseudo-dimension of $([T_i], 1, \mathcal{P}_\lambda, \psi_i)$ over weight functions $u : [N] \to \mathbb{R}_{\geq 0}$ is at most $O(d^8)$.*

*Proof.* The argument is almost the same as in Lemma A.9. The parameter space of $\psi_i$ consists of at most $m = O(d^2)$ parameters and $\psi_i$ can be represented by at most $l = O(d^6)$ arithmetic operations. By Lemmas A.7 and A.8, it completes the proof. $\square$

## B.2   Bounding the total sensitivity of $\psi_i$

Next, we again focus on proving $s_i$ (Line 12 of Algorithm 1) is a sensitivity function w.r.t. $\psi_i$; summarized as follows.

**Lemma B.2** ($s_i$ **is a sensitivity function for $\psi_i$**). *For each $i \in [N]$, we have that for each $t \in [T_i]$*

$$s_i(t) \geq \max_{\mu \in \mathbb{R}^d, \Lambda \in \mathcal{P}_{\tau,\lambda}} \frac{\psi_{it}(\mu, \Sigma, \Lambda)}{\psi_i(\mu, \Sigma, \Lambda)}.$$

*Moreover, $\sum_{t \in [T_i]} s_i(t) = O(D/\lambda)$.*

Similar to Section A, we introduce another function $s_i^{(O)} : [T_i] \to \mathbb{R}_{\geq 0}$ as a sensitivity function w.r.t. $\psi_i^{(O)}$, i.e., for any $t \in [T_i]$,

$$s_i^{(O)}(t) := \max_{\mu \in \mathbb{R}^d} \frac{\psi_{it}^{(O)}(\mu)}{\psi_i^{(O)}(\mu)}.$$

Define $\mathcal{G}_i^{(O)} := \sum_{t \in [T_i]} s_i^{(O)}(t)$ to be the total sensitivity w.r.t. $\psi_i^{(O)}$. We first have the following lemma, whose proof idea is simply from Lemma A.11 and [37, Lemma 4.4].

**Lemma B.3** (**Relation between sensitivities w.r.t. $\psi_{it}^{(O)}$ and $\psi_{it}$**). *For each $t \in [T_i]$, we have*

$$s_i(t) \leq 4D\lambda^{-1} \cdot \left( s_i^{(O)}(t) + \sum_{j=1}^{\min\{t-1,1\}} s_i^{(O)}(t-j) \right).$$

*Proof.* By the same argument as in Lemma A.11, we have that

$$s_i(t) \leq D\lambda^{-1} \cdot \max_{\mu \in \mathbb{R}^d, \Lambda \in \mathcal{D}_\lambda^d} \frac{\psi_{it}(\mu, I_d, \Lambda)}{\psi_i^{(O)}(\mu)}.$$

By a similar argument as in [37, Lemma 4.4], we have that

$$\psi_{it}(\mu, I_d, \Lambda) \leq \psi_{it}^{(O)}(\mu) + \sum_{j=1}^{\min\{t-1,1\}} \psi_{i,t-q}^{(O)}(\mu).$$

Combining the above two inequalities, we complete the proof. $\square$

Then we have the following lemma that relates $s_i^{(O)}$ and $s_i^c$ (Line 11 of Algorithm 1), whose proof follows from [57, Theorem 7] for the case that $k = 1$.

**Lemma B.4 (Sensitivities w.r.t. $\psi_i^{(O)}$).** *For each $i \in [N]$, the following holds:*

*1. For each $t \in [T_i]$, we have*

$$s_i^{(O)}(t) \leq s_i^c(t).$$

*2. $\mathcal{G}_i^{(O)} \leq 8$.*

Note that Lemma B.2 is a direct corollary of Lemmas B.3 and B.4.

## C Proof of Theorem 4.3

*Proof.* Note that the coreset size $|S| = ML$ matches the bound in Theorem 4.3. We first prove the correctness. For any $i \in I_S$, we have

$$- \ln \sum_{l \in [k]} \alpha_l \cdot \exp(-\frac{1}{2T_i} \sum_{t \in J_{S,i}} w^{(i)}(t) \cdot \psi_{it}(\mu^{(l)}, \Sigma^{(l)}, \Lambda^{(l)}))$$

$$\leq \quad - \ln \sum_{l \in [k]} \alpha_l \cdot \exp(-\frac{1}{2T_i}(1 + \varepsilon) \cdot \psi_i(\mu^{(l)}, \Sigma^{(l)}, \Lambda^{(l)})) \qquad \text{(Lemma 4.5)}$$

$$\leq \quad - \ln \sum_{l \in [k]} \left( \alpha_l \cdot \exp(-\frac{1}{2T_i}\psi_i(\mu^{(l)}, \Sigma^{(l)}, \Lambda^{(l)})) \right)^{1+\varepsilon} \qquad (x^{1+\varepsilon} \text{ is convex when } x \in [0, 1])$$

$$\leq \quad (1 + \varepsilon) \cdot \left( - \ln \sum_{l \in [k]} \alpha_l \cdot \exp(-\frac{1}{2T_i}\psi_i(\mu^{(l)}, \Sigma^{(l)}, \Lambda^{(l)})) \right).$$

Symmetrically, we can also verify that

$$- \ln \sum_{l \in [k]} \alpha_l \cdot \exp(-\frac{1}{2T_i} \sum_{t \in J_{S,i}} w^{(i)}(t) \cdot \psi_{it}(\mu^{(l)}, \Sigma^{(l)}, \Lambda^{(l)}))$$

$$\geq \quad (1 - \varepsilon) \cdot \left( - \ln \sum_{l \in [k]} \alpha_l \cdot \exp(-\frac{1}{2T_i}\psi_i(\mu^{(l)}, \Sigma^{(l)}, \Lambda^{(l)})) \right).$$

Consequently, we have

$$- \ln \sum_{l \in [k]} \alpha_l \cdot \exp(-\frac{1}{2T_i} \sum_{t \in J_{S,i}} w^{(i)}(t) \cdot \psi_{it}(\mu^{(l)}, \Sigma^{(l)}, \Lambda^{(l)}))$$

$$\in \quad (1 \pm \varepsilon) \cdot \left( - \ln \sum_{l \in [k]} \alpha_l \cdot \exp(-\frac{1}{2T_i}\psi_i(\mu^{(l)}, \Sigma^{(l)}, \Lambda^{(l)})) \right). \qquad (8)$$

Combining with Lemma 4.4, we have that

$$f'_S(\alpha, \theta)$$

$$= \quad - \sum_{i \in I_S} w(i) \cdot \ln \sum_{l \in [k]} \alpha_l \cdot \exp(-\frac{1}{2T_i} \sum_{t \in J_{S,i}} w^{(i)}(t) \cdot \psi_{it}(\mu^{(l)}, \Sigma^{(l)}, \Lambda^{(l)}))$$

$$\in \quad (1 \pm \varepsilon) \cdot \left( - \sum_{i \in I_S} w(i) \cdot \ln \sum_{l \in [k]} \alpha_l \cdot \exp(-\frac{1}{2T_i}\psi_i(\mu^{(l)}, \Sigma^{(l)}, \Lambda^{(l)})) \right) \qquad \text{(Ineq. (8))}$$

$$\in \quad (1 \pm \varepsilon)^2 \cdot f'(\alpha, \theta). \qquad \text{(Lemma 4.4)}$$

By replacing $\varepsilon$ with $O(\varepsilon)$, we prove the correctness.

For the computation time, the computation in Line 2 costs $O(d \sum_{i \in [N]} T_i)$ time since each $b_i$ and $a_i$ can be computed in $O(dT_i)$ time, and $A$ can be computed in $O(N)$ time. In Line 3, it costs $O(Ndk \ln N \ln k)$ time to solve the $k$-means clustering problem by $k$-means++. Line 4 costs $O(Ndk)$ time since each $p(i)$ costs $O(dk)$ time to compute. Lines 5-6 cost $O(Nd)$ time for computing sensitivity function $s$. Lines 7-8 cost $O(N)$ time for constructing $I_S$. Overall, it costs $O(d \sum_{i \in [N]} T_i + Ndk \ln N \ln k)$ at the first stage. Line 10 costs at most $O(dT_i)$ time to compute $\mathsf{OPT}_i^{(O)}$. Lines 11-12 cost $O(T_i)$ time to compute $s_i$. Lines 13-14 cost $O(T_i)$ time to construct $J_{S,i}$. Since $|I_S| \le N$, we have that it costs at most $O(d \sum_{i \in [N]} T_i)$ time at the second stage. We complete the proof. □

## D   Discussion on lower bounds

There is no provable lower bound result for our GMM coreset with time series data. We conjecture that without the first condition in Assumption 4.1, the coreset size should depend exponentially in $k$ and logarithmic in $n$. The motivation is from a simple setting that all $T_i = 1$ (GMM with static data), in which [26] reduces the problem to projective clustering whose coreset size depends exponentially in $k$ and logarithmic in $n$. Moreover, [26] believes that these dependencies are unavoidable for GMM coreset.