# OpenReview forum: "Coresets for Time Series Clustering"
_NeurIPS.cc/2021/Conference — NeurIPS 2021 Spotlight_

### Official Review · Reviewer_eQJc · 2021-07-07

**Rating:** 6
**Confidence:** 2

**Summary:**

The authors propose a novel coreset construction framework for the problem of clustering multivariate time series. To my knowledge, this is the first construction of coresets in this setting. The framework, as well as the proof scheme, is close to the work in [45], in which Lucic et al. studied the problem of coreset constructions for the MLE task under the GMM model.

Here, the authors add to the GMM model a temporality and thus an additional autocorrelation in the underlying model. The proof's scheme is similar to that of [45] in the sense that they also reduce the GMM time series clustering problem to a $k$-means clustering problem (and then use more classical results for $k$-means from the state-of-the-art).

**Main Review:**

The additional temporal dimension compared to GMMs makes the coreset algorithm and the proofs of the theorems more involved. Given the information provided in the proof sketches (I did not read the supplementary material), I did not find any error in the reasoning or computation; but I could not guarantee that there are no errors.

The idea of first sampling entities, and then a subset of time periods for each selected entities is quite natural. The way I see it, the first stage of the algorithm is very close to the work in [45], with a k-means++ based routine to estimate upper bounds of the k-means' sensitivity. Then, the second part of the algorithm (selecting time periods for each sampled entity) is close to the work in [34]. This work is a mix of ideas from [45] and [34].

I am frustrated by the experimental section in which you only compare your work with uniform sampling.
- What if you keep all entities but sample only time? What if you keep all time periods but sample only entities?
- What if you forget the time in your time series and directly apply the algorithm from [45]? What gain do you truly get with your approach?

**Time Spent Reviewing:**

5

---

> ### Author Response · Authors · 2021-08-11
> **Thanks for your positive review and feedback. We answer your specific questions below and hope that you will increase your support for our paper.**
>
> > **"What if you keep all entities but sample only time? What if you keep all time periods but sample only entities?"**
>
> *Response:* These two approaches will result in a subset of entity-time pairs of size dependent on either the number of entities $N$ or the number of time periods $T$, which is much larger than our coreset size. However, empirically, we want to compare the achieved likelihood ratio under the same size between our coreset and baselined algorithms. Therefore, we did not include these two approaches as our baselines.
>
> Please also note that in response to your next comment, we have used coresets for cross-sectional data that ignores the time series aspect as a new baseline and compare it with different levels of auto-correlations.
>
> We hope you agree with our reasoning for why we chose the baselines we have.
>
> > **"What if you forget the time in your time series and directly apply the algorithm from [45]? What gain do you truly get with your approach?"**
>
> *Response:* To capture the spirit of your comment (and feedback from other review team members), we now compare against a coreset algorithm for cross-sectional data where $T$=1,(specifically [23]) which ignores the time series aspects.
>
> The results show that our coreset for time series clustering outperforms the coreset for cross-section clustering in [23], both in terms of fit ($\gamma_S$) and computation time. We hope this further clarifies the empirical value of the time series clustering coreset relative  to the existing literature, which only has developed clustering coresets for cross-sectional data.
>
>
> **Table 1a: Coreset performance (fit) on Synthetic Data 1 ($N=T=500$).**
>
> | $\varepsilon$ | $V_S^\star$ of **CRGMM** | $V_S^\star$ of **Uni** | $V_S^\star$ of **LFKF** | $V^\star$ | $\gamma_S^\star$ of **CRGMM** | $\gamma_S^\star$ of **Uni** | $\gamma_S^\star$ of **LFKF** |   size  |
> | ------------- | ------------------------ | ---------------------- | ----------------------- | --------- | ----------------------------- | --------------------------- | ---------------------------- | --- |
> | 0.1           |      2050                    |        2058                |            2069             |   2041  |  22   |   34  |  56  |  1514
> |   0.2       |     2093     |     2210     |   2264  |  2041   |   104  |   342  |   446  | 404
> |   0.3       |     2194     |     2398     |  2384   |  2041   |   306  |   714  |  686   | 191
> |   0.4       |     2335     |      3963    |   2705  |   2041  |   588  |   3844  |  1328   | 93
> | 0.5     |   2383   |   3304   |  3461   |  2041   |   684  |   2526  |   2840  | 72
>
> **Table 1b: Coreset performance (fit) on Synthetic Data 2 ($N=200, T=1250$).**
>
> | $\varepsilon$ | $V_S^\star$ of **CRGMM** | $V_S^\star$ of **Uni** | $V_S^\star$ of **LFKF** | $V^\star$ | $\gamma_S^\star$ of **CRGMM** | $\gamma_S^\star$ of **Uni** | $\gamma_S^\star$ of **LFKF** |  size   |
> | ------------- | ------------------------ | ---------------------- | ----------------------- | --------- | ----------------------------- | --------------------------- | ---------------------------- | --- |
> | 0.1           |         811                |       825                 |              841           |     812      |   2                            |        26                     |                 58             |   1718  |
> | 0.2           |          824                |       895                 |               871          |      812     |                24               |          166                   |             118                 |   447  |
> | 0.3           |           864               |      958                  |           994              |     812      |                  104             |           292                  |             364                 |   199  |
> | 0.4           |       860                   |      1190                  |              1114           |      812     |               96                |          756                   |               604               |   98  |
> | 0.5           |          910                |        1361                |            1284             |      812     |               196                |           1098                  |               944               |  71   |
>
>
> **Table 2a: Coreset performance (computation time) on Synthetic Data 1 ($N=T_i=500$).**
>
> | $\varepsilon$ | $T_S+T_C$(s) of **CRGMM** | $T_S+T_C$(s) of **Uni** | $T_S+T_C$(s) of **LFKF** | $T_\mathcal{X}$ |
> | ------------- | ------------------------- | ----------------------- | ------------------------ | :--- |
> | 0.1           |           109                |     2416                    |             1355             |   3436  |
> | 0.2           |            41               |    1054                     |             652             |  3436   |
> | 0.3           |            62               |     419                    |                392          |   3436  |
> | 0.4           |             38              |    621                     |         260                 |   3436  |
> | 0.5           |          47                 |    132                     |              147            |  3436   |
>
> **Table 2b: Coreset performance (computation time) on Synthetic Data 2 ($N=200$, $T_i=1250$).**
>
> | $\varepsilon$ | $T_S+T_C$(s) of **CRGMM** | $T_S+T_C$(s) of **Uni** | $T_S+T_C$(s) of **LFKF** | $T_\mathcal{X}$ |
> | ------------- | ------------------------- | ----------------------- | ------------------------ | :--- |
> | 0.1           |         694                  |     1859                    |             1687             |   9787  |
> | 0.2           |            139               |     1991                    |             1484             |   9787  |
> | 0.3           |             51              |     527                    |               832           |   9787  |
> | 0.4           |             43              |    408                     |                 450         |  9787   |
> | 0.5           |              57             |     389                    |              277            |  9787   |

---

> > ### Comment · Reviewer_eQJc · 2021-08-23
> > **thank you for your response**
> >
> > thank you for your response. The added experiment helps grasp the novelty of your approach

---

### Official Review · Reviewer_V91b · 2021-07-10

**Rating:** 6
**Confidence:** 4

**Summary:**

The paper considers the problem of clustering time series data. The authors provide a coreset for this problem, under some assumptions. The coreset size depends polynomially on the number of clusters k, the dimension d, and 1/\varepsilon and the parameters related to the assumption on the data, and requires near-linear time to compute.
The authors validate their coreset empirically on synthetic data, as compared to a uniform random sample. They show that the coreset: (i) achieves better results can the random sample, and (i) can help accelerate the computational time, relative to the full data, by 1-2 orders of magnitude, with only a small compromise in the accuracy.

**Limitations And Societal Impact:**

Yes.

**Main Review:**

Pros:
- The problem is indeed very interesting and relevant.
- The paper is well organized.
- The overviews provided are very helpful, as well as the detailed comparison of Theorem 4.3 to the related previous works.
- In overall, I really like the paper and its results.

Cons:
- The contributions paragraph is very high-level, and lacks the important details (e.g., coreset size, actual computation time, etc.).
- The coreset obtains an additive error of epsilon times f', where f' is a different cost function. This relation should be investigated more thoroughly, both in theory and in the conducted experiments.
- While the analysis and proofs are indeed deep and non-trivial, the novelty of the results is not fully clear; It is unclear if this is the first coreset for such time-series data, and unclear whether existing works, which handle the same cost function but with static data, can be easily adapted to the case of time series data. This concern arises since, eventually, the provided coreset for time series data is basically a union between a coreset for entity level data (ignoring the time-series data for each entity), and a separate coreset for each entity. Hence, one might think that the novelty of the paper lies mostly in the successful decoupling of the time-series problem into two sub-problems with static data. I would recommend that the authors add a "hardness" paragraph that discusses this topic.
- The hardness of computing a coreset for time series clustering is not discussed. Is there a provable lower bound for the coreset size? This can better justify the coreset sizes obtained in the paper.
- The problem seems, in some sense, related to the problem of clustering input sets, or clustering shapes, as e.g.,
(i) “Sets Clustering”, Jubran et al.
(ii) "k-Means Clustering of Lines for Big Data", Merom and Feldman.
(iii) Works that handle probabilistic databases (for example "Smallest enclosing ball for probabilistic data" by Munteanu et al.
Establishing a connection to those problems seems very relevant.
- Regarding the experiments:
(i) how was the optimization problem solved on the full data? was it done similarly to the EM solution applied on the coreset?
(ii) It seems a bit strange that the the proposed coreset achieves a 2x-8x acceleration in solving Problem 3.1 compared to the uniform sampling, as the random sample is computed much faster, and is of the same size as the corese. Can the authors elaborate on this?
(iii) The dimension d tested was only d=2 (probably due to the large dependency, in theory, on d).
However, as the theory in most coreset papers is far too pessimistic, it would be interesting to see this effect in practice for d>2.


Other detailed comments:
- There are some typos / writing issues / undefined variables. For example:
(i) In the definition of the input set X, the subscript of the first element x_{i1} of every Xi does not have a comma, while the last element x_{i,Ti} does have a comma. It should be consistent. Also, Xi is said to be a subset of R^{Ti x d}. However, as Xi is not a set, the "\subset" should be replaced with "\in", and it should be R^{d\times Ti} as there are Ti columns and d rows in Xi.
(ii) The argmax in Line 37: it is unclear over what set are we maximizing the expression.
(iii) Line 38: missing closing bracket.
(iv) Line 80: "where objective" -> "where the objective".
(v) Line 91: "compared to uniform", sentence not complete.
(vi) Line 174: If i'm not missing anything, then P is undefined (should be P_X). Similarly in Definition 3.3.
(vii) Line 182: Extra closing bracket.
- The organization of the column titles in Table 1 are very confusing. I would recommend either spacing the titles more, or placing some vertical dividers in between unrelated columns.
- An illustration figure to demonstrate the "GMM clustering with time series data" problem (even for d=2) would be very appreciated.

I am willing to re-evaluate my score after addressing the above concerns.

**Time Spent Reviewing:**

5

---

> ### Author Response · Authors · 2021-08-11
> **Thanks for your positive review and feedback. We answer your specific questions below and hope that you will increase your support for our paper.**
>
> > **"The contributions paragraph is very high-level, and lacks the important details (e.g., coreset size, actual computation time, etc.)."**
>
> *Response:* Thanks for your feedback and we will add these details to the contribution paragraph.
>
> > **"The coreset obtains an additive error of epsilon times $f'$, where $f'$ is a different cost function. This relation should be investigated more thoroughly, both in theory and in the conducted experiments."**
>
> *Response:* Thanks for your suggestion and we will add a discussion on the relation between $f’_S$ and $f$. In theory, as a similar argument as Inequality (2), we have that $f'_S(\alpha'(\theta), \theta) + \phi(\alpha, \theta) \in (1\pm \varepsilon)\cdot f'(\alpha,\theta) + \phi(\alpha,\theta) \in (1\pm \varepsilon)\cdot f(\alpha, \theta) \pm 2\varepsilon \phi(\alpha,\theta)$. Thus, if $\phi(\alpha, \theta)\lesssim f(\alpha, \theta)$, we conclude that $f'_S(\alpha'(\theta), \theta)+ \phi(\alpha,\theta) \in (1\pm 3\varepsilon)\cdot f(\alpha, \theta)$. For the empirical results, we will add a comparison between optimizing $f_S(\alpha, \theta)$ directly and optimizing $f'_S(\alpha'(\theta), \theta) + \phi(\alpha, \theta)$ on the coreset $S$ in the final version.
>
> > **"I would recommend that the authors add a "hardness" paragraph that discusses this topic."**
>
> *Response:* Thank you for the suggestion and we will add a paragraph to discuss our technical novelty. While we agree that our entity coreset uses some ideas from [23], novel technical ideas are required in our construction. A simple generalization of [23] to time series data is to treat all observations $x_{it}$ independent and compute a sensitivity for each $x_{it}$ directly for importance sampling. However, this idea can not capture the property that multiple observations $x_{i,1},\ldots,x_{i,T_i}$ are drawn from the same Gaussian model (certain $l\in [k]$). To handle multiple observations, we show that although each $\psi_i$ consists of $T_i$ sub-functions $\psi_{it}$, it can be approximated by a single function on the average observation $b_i = \frac{\sum_{t\in [T_i]}x_{it}}{T_i}$, specifically, we have $\psi_i(\mu, I_d, 0_d) = d(b_i, \mu)^2 + O(1)$. This property enables us to "reduce" the representative complexity of $\psi_i$, and motivate two key steps of our construction: 1) For the **sensitivity function**, we reduce to a certain $k$-means clustering problem on average observations $b_i = \frac{\sum_{t\in [T_i]}x_{it}}{T_i}$ of entities (Definition 4.2) by upper bounding the maximum effect of covariances and autocorrelations and applying the fact that $\psi_i(\mu, I_d, 0_d) = d(b_i, \mu)^2 + O(1)$. 2) For the **pseudo-dimension**, we prove that there are only $\mathrm{poly}(k,d)$ intristic operators in $\psi_i$ between parameters $\theta$ and observations $x_{it}$, based on the reduction of the representative complexity of $\psi_i$.
>
> > **"Is there a provable lower bound for the coreset size?"**
>
> *Response:* No, we do not have a provable lower bound for the coreset size. We conjecture that without the assumption on the condition number ((1) in Assumption 4.1), the coreset size should depend exponentially in $k$ and logarithmic in $n$. The motivation is from a simple setting that all $T_i=1$ (GMM with static data), in which [25] reduces the problem to projective clustering whose coreset size depends exponentially in $k$ and logarithmic in $n$. Moreover, [25] believes that these dependencies are unavoidable for GMM coreset. Investigating this direction is an interesting topic for future work.
>
> > **"Establishing a connection to those problems (clustering input sets, or clustering shapes) seems very relevant."**
>
> *Response:* Thank you for the suggestion. We will add a paragraph to discuss these problems in the related work section in the final version.
>
> > **"How was the optimization problem solved on the full data? Was it done similarly to the EM solution applied on the coreset?"**
>
> *Response:* Yes. As mentioned in Footnote 2, we implement an EM algorithm on both the full data and the coreset.
>
> > **"It seems a bit strange that the proposed coreset achieves a 2x-8x acceleration in solving Problem 3.1 compared to the uniform sampling, as the random sample is computed much faster, and is of the same size as the coreset. Can the authors elaborate on this?"**
>
> *Response:* By Table 1, we can see that the construction time $T_C$ is at most $2$ seconds for both our coreset and uniform sampling, which is at most 5\% of the computation time $T_\mathcal{X}$. Thus, our coreset achieves a 2x-8x acceleration in solving Problem 3.1 compared to the uniform sampling since the computation time on coreset is much faster. As mentioned in Footnote 3, a possible explanation is that our coreset selects $I_S$ of entities that are more representative than uniform sampling, and hence, achieves a better convergence speed.
>
> > **"The dimension d tested was only $d=2$ (probably due to the large dependency, in theory, on $d$). However, as the theory in most coreset papers is far too pessimistic, it would be interesting to see this effect in practice for $d>2$."**
>
> *Response:* We agree with your point that it would be interesting to assess sensitivity of the coreset size
> $d$ in practice. However, given competing suggestions for additional testing, we decided to prioritize on the suggestion to compare our coreset for time series clustering against the coreset for cross-sectional clustering based on [23]. We find that our coreset outperforms the coreset from [23] on both fit and computation time.
>
> > **"In the definition of the input set $X$, the subscript of the first element $x_{i1}$ of every $X_i$ does not have a comma, while the last element $x_{i,Ti}$ does have a comma. It should be consistent. Also, $X_i$ is said to be a subset of $\mathbb{R}^{T_i \times d}$. However, as $X_i$ is not a set, the "$\subset$" should be replaced with "$\in$", and it should be $\mathbb{R}^{d\times T_i}$ as there are $T_i$ columns and $d$ rows in $X_i$."**
>
> *Response:* We will make $x_{i,t}$ consistent and fix these typos.
>
> > **"The argmax in Line 37: it is unclear over what set are we maximizing the expression."**
>
> *Response:* It should be $\arg\max_{\alpha, \theta^{(1)}, \ldots, \theta^{(k)}}$.
>
> > **"Line 38: missing closing bracket."**
>
> *Response:* Yes, it should be $\theta^{(l)})$ instead of $\theta^{(l)}$.
>
> > **"Line 80: "where objective" -> "where the objective"."**
>
> *Response:* Yes, it is a typo.
>
> > **"Line 91: "compared to uniform", sentence not complete."**
>
> *Response:* It should be "uniform sampling" instead of "uniform".
>
> > **"Line 174: If I'm not missing anything, then $P$ is undefined (should be $P_\mathcal{X}$). Similarly in Definition 3.3."**
>
> *Response:* Yes, $P$ should be $P_\mathcal{X}$.
>
> > **"Line 182: Extra closing bracket."**
>
> *Response:* Yes, it should be $\Sigma^{(l)}$ instead of $\Sigma^{(l)})$.
>
> > **"I would recommend either spacing the titles more, or placing some vertical dividers in between unrelated columns."**
>
> *Response:* Thank you for the suggestion. We will place vertical dividers in between unrelated columns.
>
> > **"An illustration figure to demonstrate the "GMM clustering with time series data" problem (even for d=2) would be very appreciated."**
>
> *Response:* Thank you for the suggestion. We will include an illustration figure in the final version.

---

### Official Review · Reviewer_3r9a · 2021-07-12

**Rating:** 7
**Confidence:** 3

**Summary:**

The paper suggests a coreset construction algorithm for a clustering time-series data.


**Limitations And Societal Impact:**


The authors have addressed the limitations of the paper properly.
Specifically speaking, the coreset construction algorithm is performed well under a mild boundedness assumption on the covariance matrices of the underlying Gaussians.

The coreset size on the other hand depends polynomially on the dimension of the instances, i.e., the exponent is $16$. Thus in such cases, when dealing with big data where $d \in \Theta{M}$ or even $d \in \Theta{M^{1/16}}$, then the coreset basically contains $M$ points where $M$ denotes the total number of instances in the dataset.

Please provide a small discussion regarding the lower bound on the coreset size which is needed for GMMs on time-series data, i.e., "how hard the problem is from a coreset point of view?"



**Main Review:**

In this paper, a coreset construction algorithm was provided for clustering time-series data, specifically speaking GMMs.
The coreset was achieved via reduction towards the $k$-means clustering problem.

In addition, I find it interesting that the paper suggests two different weight functions, one for the entities and one for the time periods whereas for traditional coresets, there is only one weight function that handles the only entities.

I enjoyed reading the paper, however, due to time constraints, I was not able to go over the proofs of the presented theorems.

Comments:
- Line 141, $\Delta_k$ instead of $\Delta^k$.
- $e_{it}$ is not a scalar based on its definition at Line 150. Did you mean to write $e_{it} \in \mathbb{R}^{d}$?
- Why didn't you apply your coreset on real-world panel data?
- The coreset size depends heavily on the dimension of the data. See the limitation section for elaborated improvements.




**Time Spent Reviewing:**

5 hours

---

> ### Author Response · Authors · 2021-08-11
> **Thank you for your positive review and feedback. We answer your specific questions below.**
>
> > **"Line 141, $\Delta_k$  instead of $\Delta^k$"**
>
> *Response:* Yes, you are right. It is a typo.
>
> > **"$e_{it}$ is not a scalar based on its definition at Line 150. Did you mean to write $e_{it}\in\mathbb{R}^d$?"**
>
> *Response:* Yes, we mean $e_{it}\in\mathbb{R}^d$.
>
> > **"Why didn't you apply your coreset on real-world panel data?"**
>
> *Response:* It would indeed be a valuable future direction to empirically evaluate our findings on real-world datasets. But given space constraints and the objective of our current paper to show the value of coresets for time series clustering, we chose to further elaborate on the value of our coreswet with respect to add an additional benchmark using coresets for cross-sectional data (i.e., $T$=1, where entity-time pairs are considered independent; specifically we use [23]). We discuss the details of those results below. We hope you agree with our prioritization.
>
> The results show that our coreset for time series clustering outperforms the coreset for cross-section clustering in [23], both in terms of fit ($\gamma_S$) and computation time. We hope this further clarifies the empirical value of the time series clustering coreset relative  to the existing literature, which only has developed clustering coresets for cross-sectional data.
>
>
> **Table 1a: Coreset performance (fit) on Synthetic Data 1 ($N=T=500$).**
>
> | $\varepsilon$ | $V_S^\star$ of **CRGMM** | $V_S^\star$ of **Uni** | $V_S^\star$ of **LFKF** | $V^\star$ | $\gamma_S^\star$ of **CRGMM** | $\gamma_S^\star$ of **Uni** | $\gamma_S^\star$ of **LFKF** |   size  |
> | ------------- | ------------------------ | ---------------------- | ----------------------- | --------- | ----------------------------- | --------------------------- | ---------------------------- | --- |
> | 0.1           |      2050                    |        2058                |            2069             |   2041  |  22   |   34  |  56  |  1514
> |   0.2       |     2093     |     2210     |   2264  |  2041   |   104  |   342  |   446  | 404
> |   0.3       |     2194     |     2398     |  2384   |  2041   |   306  |   714  |  686   | 191
> |   0.4       |     2335     |      3963    |   2705  |   2041  |   588  |   3844  |  1328   | 93
> | 0.5     |   2383   |   3304   |  3461   |  2041   |   684  |   2526  |   2840  | 72
>
> **Table 1b: Coreset performance (fit) on Synthetic Data 2 ($N=200, T=1250$).**
>
> | $\varepsilon$ | $V_S^\star$ of **CRGMM** | $V_S^\star$ of **Uni** | $V_S^\star$ of **LFKF** | $V^\star$ | $\gamma_S^\star$ of **CRGMM** | $\gamma_S^\star$ of **Uni** | $\gamma_S^\star$ of **LFKF** |  size   |
> | ------------- | ------------------------ | ---------------------- | ----------------------- | --------- | ----------------------------- | --------------------------- | ---------------------------- | --- |
> | 0.1           |         811                |       825                 |              841           |     812      |   2                            |        26                     |                 58             |   1718  |
> | 0.2           |          824                |       895                 |               871          |      812     |                24               |          166                   |             118                 |   447  |
> | 0.3           |           864               |      958                  |           994              |     812      |                  104             |           292                  |             364                 |   199  |
> | 0.4           |       860                   |      1190                  |              1114           |      812     |               96                |          756                   |               604               |   98  |
> | 0.5           |          910                |        1361                |            1284             |      812     |               196                |           1098                  |               944               |  71   |
>
>
> **Table 2a: Coreset performance (computation time) on Synthetic Data 1 ($N=T_i=500$).**
>
> | $\varepsilon$ | $T_S+T_C$(s) of **CRGMM** | $T_S+T_C$(s) of **Uni** | $T_S+T_C$(s) of **LFKF** | $T_\mathcal{X}$ |
> | ------------- | ------------------------- | ----------------------- | ------------------------ | :--- |
> | 0.1           |           109                |     2416                    |             1355             |   3436  |
> | 0.2           |            41               |    1054                     |             652             |  3436   |
> | 0.3           |            62               |     419                    |                392          |   3436  |
> | 0.4           |             38              |    621                     |         260                 |   3436  |
> | 0.5           |          47                 |    132                     |              147            |  3436   |
>
> **Table 2b: Coreset performance (computation time) on Synthetic Data 2 ($N=200$, $T_i=1250$).**
>
> | $\varepsilon$ | $T_S+T_C$(s) of **CRGMM** | $T_S+T_C$(s) of **Uni** | $T_S+T_C$(s) of **LFKF** | $T_\mathcal{X}$ |
> | ------------- | ------------------------- | ----------------------- | ------------------------ | :--- |
> | 0.1           |         694                  |     1859                    |             1687             |   9787  |
> | 0.2           |            139               |     1991                    |             1484             |   9787  |
> | 0.3           |             51              |     527                    |               832           |   9787  |
> | 0.4           |             43              |    408                     |                 450         |  9787   |
> | 0.5           |              57             |     389                    |              277            |  9787   |
>
>
>
>
> > **"Please provide a small discussion regarding the lower bound on the coreset size which is needed for GMMs on time-series data."**
>
> *Response:* Thank you for the comment and we will add a discussion regarding the lower bound on the coreset size in the final version. There is no provable lower bound result. We conjecture that without the assumption on the condition number ((1) in Assumption 4.1), the coreset size should depend exponentially in $k$ and logarithmic in $n$. The motivation is from a simple setting that all $T_i=1$ (GMM with static data), in which [25] reduces the problem to projective clustering whose coreset size depends exponentially in $k$ and logarithmic in $n$. Moreover, [25] believes that these dependencies are unavoidable for GMM coreset.

---

### Official Review · Reviewer_xadL · 2021-07-16

**Rating:** 5
**Confidence:** 4

**Summary:**

This paper uses the Feldman Langberg sensitivity framework to compute coresets for time series clustering under a Gaussian Mixture Model.

This imposes additional challenges since there is an instance domain as well as a time domain to approximate. The mixture model introduces covariances and to reflect the time dependence there is an additional autocorrelation introduced on the time domain.

Those covariance and correlation structures are being bounded in order to bound their impact on the sensitivities and after getting rid of all those structures the problem is reduced to a simpler clustering model for which sensitivity bounds are obtained more easily.

The resulting coresets are assessed empirically on synthetic data.

The writing is very good and the main body provides all information to understand even on a technical level, although the details are in the appendix.

In principle I would tend to accept this paper but knowing the previous literature on GMM as well as a recent coreset work on "panel data" my main impression is that the submission is just a dejavu of those two works.

**Main Review:**

My main concern for rejecting this paper is a lack of novelty and significance:

1) The model (with the assumptions imposed to get small coresets) seems inappropriate: When the covariance matrices have constantly bounded condition and the autocorrelation AR(1) is used, then every time series is just a random walk concentrated around their mean. This is very well behaved time series data.

Why is there no comparison or related work for maybe more appropriate time series (polygonal curve) clustering eg under dynamic time-warping or Frechet distance, and where values can oscillate arbitrarily. See for example the recent series of works by K. Buchin or A. Driemel or D. Rhode.

2) Technically there is not more than already seen in the papers [23] where coresets for GMMs (with bounded condition) are developed. Why is the more recent generalization without this bound [25] being dismissed? I would rather deal with more general covariances but loose the log n factor.

The submission uses the same techniques as [34] for getting rid of the autocorrelations (of order q, here only q=1 is considered). It is simply by assuming that the correlation is bounded away from 1 and by applying Cauchy-Schwarz. In [34] the two-stage coreset construction for clustering of regressions was one of the novelties but here it is just the same again.

The authors try to sell it as new or different by pointing out the two weighting functions w(i) and w_i(j) which due to non-linearities can't be combined, but in principle it was the same in [34] up to finally defining w(i,j) = w(i) * w_i(j).

another thing that they point out that the component's cost functions are not bounded as in [34] but the bound on the eigenvalues is just the same, noting that the eigenvalues are dominating the costs in the GMM.

3) Even on synthetic data the coresets show only slight improvements over uniform sampling. Since the authors are free to model the data in any way, they could at least simulate one data set that is particularly hard for uniform sampling where coresets show their strength. Currently it looks like if data follows the generating model, sensitivities are pretty uniform. Also, why is there no real-world data comparison? Is there even any real data that can be modeled similarly to the GMM model described here?

**Time Spent Reviewing:**

6

---

> ### Author Response · Authors · 2021-08-11
> **Thanks for your comments and questions that give us the chance to better clarify our novelty and contributions. We answer your questions below. We hope that they address your concerns and that you will increase your support for the paper.**
>
> > **"The model (with the assumptions imposed to get small coresets) seems inappropriate. When the covariance matrices have constantly bounded conditions and the autocorrelation AR(1) is used, then every time series is just a random walk concentrated around their mean. This is very well behaved time series data. Why is there no comparison or related work for maybe more appropriate time series (polygonal curve) clustering eg under dynamic time-warping or Frechet distance, and where values can oscillate arbitrarily. See for example the recent series of works by K. Buchin or A. Driemel or D. Rhode."**
>
> *Response:* Thank you for requesting this clarification. As we discuss in the paper, there are three key sub-literatures within time-series clustering based on whether they (i) process directly on raw data, (ii) indirectly with features extracted from the raw data, or (iii) indirectly with models built from the raw data (See [44,2] for surveys). The references you discuss are based on the first two streams of literature. Our approach is based on the third sub-literature on model-based clustering of time series data, specifically focusing on the AR(1) model. We will add a specific reference to a survey on model-based clustering to highlight the importance of this sub-literature, and its various real-world applications. (e.g., Frühwirth-Schnatter, Sylvia. "Panel data analysis: a survey on model-based clustering of time series." Advances in Data Analysis and Classification 5, no. 4 (2011): 251-280.)
>
> > **"Technically there is not more than already seen in the papers [23] where coresets for GMMs (with bounded condition) are developed."**
>
> *Response:* While we agree that our entity coreset construction uses some ideas from [23], novel technical ideas are required in our paper. A generalization of [23] to time series data is to treat all observations $x_{it}$ independent and compute a sensitivity for each $x_{it}$ directly for importance sampling. However, this idea cannot capture the property that multiple observations $x_{i,1},\ldots,x_{i,T_i}$ are drawn from the same Gaussian model (certain $l\in [k]$). To handle multiple observations, we show that although each $\psi_i$ consists of $T_i$ sub-functions $\psi_{it}$, it can be approximated by a single function on the average observation $b_i = \frac{\sum_{t\in [T_i]}x_{it}}{T_i}$, specifically, we have $\psi_i(\mu, I_d, 0_d) = d(b_i, \mu)^2 + O(1)$. This property enables us to "reduce" the representative complexity of $\psi_i$, and motivate two key steps of our construction: 1) For the **sensitivity function**, we give a reduction to a certain $k$-means clustering problem on average observations $b_i = \frac{\sum_{t\in [T_i]}x_{it}}{T_i}$ of entities (Definition 4.2) by upper bounding the maximum effect of covariances and autocorrelations and applying the fact that $\psi_i(\mu, I_d, 0_d) = d(b_i, \mu)^2 + O(1)$. 2) For the **pseudo-dimension**, we prove that there are only $\mathrm{poly}(k,d)$ intristic operators in $\psi_i$ between parameters $\theta$ and observations $x_{it}$, based on the reduction of the representative complexity of $\psi_i$. We will make the novelty over [23] more clear in the final version.
>
> > **"Why is the more recent generalization without this bound [25] being dismissed? I would rather deal with more general covariances but loose the log n factor."**
>
> *Response:* Generalizing  our results in the same manner as [25] does over [23] would be an interesting future direction. Currently, it is unclear how to generalize the approach of [25] to time series data since [25] assumes that each point is an integral point within a bounded box, while we consider a continuous GMM generative model (1), and hence, each coordinate of an arbitrary observation $x_{itr}$ is drawn from a certain continuous GMM distribution which is not integral with probability $\approx 1$ and can be unbounded. We will add this in the discussion section.
>
> > **"The authors try to sell it as new or different by pointing out the two weighting functions $w(i)$ and $w_i(j)$ which due to non-linearities can't be combined, but in principle it was the same in [34] up to finally defining $w(i,j) = w(i) * w_i(j)$."**
>
> *Response:* If you are referring to Lines 185-189, we agree with you and will clarify in the final version.
>
> > **"Another thing that they point out that the component's cost functions are not bounded as in [34] but the bound on the eigenvalues is just the same, noting that the eigenvalues are dominating the costs in the GMM."**
>
> *Response:* While the eigenvalues do affect the cost functions $\psi_i$s, the positions of means $\mu^{(l)}$s also affect $\psi_i$s. For instance, consider the case that all eigenvalues are 1 and all autocorrelations are 0, i.e., for all $l\in [k]$,  $\Sigma^{(l)}= I_d$ and $\Lambda^{(l)}=0_d$. In this case, it is easy to see that  $\frac{\max_{\mu\in \mathcal{R}^d}\psi_i(\mu, I_d, 0_d)}{\min_{\mu\in \mathcal{R}^d}\psi_i(\mu, I_d, 0_d)} = \infty$, i.e., the value of $\psi_i(\mu, I_d, 0_d)$ is unbounded as $\mu$ changes. Differently, the component's cost functions are bounded in [34] since $\mu^{(l)}$s do not appear in [34]. We will clarify this point in the final version.
>
> > **"Even on synthetic data the coresets show only slight improvements over uniform sampling."**
> >
> > *and*
> >
> >
> > **"Since the authors are free to model the data in any way, they could at least simulate one data set that is particularly hard for uniform sampling where coresets show their strength. Currently it looks like if data follows the generating model, sensitivities are pretty uniform."**
>
> *Response to the above two comments:*
>
> As mentioned at Line 89, to achieve a similar fit with the full data, our coreset needs fewer entity-time observations (<6%); therefore, the computation time for a given level of accuracy reduces by a 2x-8x factor when compared to uniform.
>
>
> Given our clarification above that our coreset does indeed do better than uniform sampling, we hope you agree that that the existing simulated datasets already show the strength of our coresets.
>
>
> > **"Why is there no real-world data comparison? Is there even any real data that can be modeled similarly to the GMM model described here?"**
>
> *Response:*
>
> *Can real data be modeled using our GMM model?* As we discussed earlier, model-based clustering of time series data is an important sub-literature in time series clustering; and AR(1) models with 1 period lag are a primary workhorse model for time series data modeling. In the model-based clustering survey paper mentioned above, please see Eq (1) in Section 2.1 that reflects the first-order Markov Process captured by our AR(1) model.
>
> *Why no real-world data comparison?* It would indeed be a valuable future direction to empirically evaluate our findings on real-world datasets. But given space constraints and the objective of our current paper to show the value of coresets for time series clustering, we chose to further elaborate on the value of our coreset with respect to adding an additional benchmark using coresets for cross-sectional data (i.e., $T$=1), where entity-time pairs are considered independent; specifically we use [23]). We discuss the details of those results below. We hope you agree with our prioritization.
>
> The results show that our coreset for time series clustering outperforms the coreset for cross-section clustering in [23], both in terms of fit ($\gamma_S$) and computation time. We hope this further clarifies the empirical value of the time series clustering coreset relative to the existing literature, which only has developed clustering coresets for cross-sectional data.
>
>
> **Table 1a: Coreset performance (fit) on Synthetic Data 1 ($N=T=500$).**
>
> |$\varepsilon$|$V_S^\star$ of **CRGMM**|$V_S^\star$ of **Uni**|$V_S^\star$ of **LFKF**|$V^\star$|$\gamma_S^\star$ of **CRGMM**|$\gamma_S^\star$ of **Uni**|$\gamma_S^\star$ of **LFKF**|size|
> | ------------- | ------------------------ | ---------------------- | ----------------------- | --------- | ----------------------------- | --------------------------- | ---------------------------- | --- |
> |0.1|2050|2058|2069|2041|22|34|56|1514
> |0.2|2093|2210|2264|2041|104|342|446|404
> |0.3|2194|2398|2384|2041|306|714|686|191
> |0.4|2335|3963|2705|2041|588|3844|1328|93
> |0.5|2383|3304|3461|2041|684|2526|2840|72
>
> **Table 1b: Coreset performance (fit) on Synthetic Data 2 ($N=200, T=1250$).**
>
> |$\varepsilon$|$V_S^\star$ of **CRGMM**|$V_S^\star$ of **Uni**|$V_S^\star$ of **LFKF**|$V^\star$|$\gamma_S^\star$ of **CRGMM**|$\gamma_S^\star$ of **Uni**|$\gamma_S^\star$ of **LFKF**|size|
> | ------------- | ------------------------ | ---------------------- | ----------------------- | --------- | ----------------------------- | --------------------------- | ---------------------------- | --- |
> |0.1|811|825|841|812|2|26|58|1718|
> |0.2|824|895|871|812|24|166|118|447|
> |0.3|864|958|994|812|104|292|364|199|
> |0.4|860|1190|1114|812|96|756|604|98|
> |0.5|910|1361|1284|812|196|1098|944|71|
>
>
> **Table 2a: Coreset performance (computation time) on Synthetic Data 1 ($N=T_i=500$).**
>
> |$\varepsilon$|$T_S+T_C$(s) of **CRGMM**|$T_S+T_C$(s) of **Uni**|$T_S+T_C$(s) of **LFKF**|$T_\mathcal{X}$|
> | ------------- | ------------------------- | ----------------------- | ------------------------ | :--- |
> |0.1|109|2416|1355|3436|
> |0.2|41|1054|652|3436|
> |0.3|62|419|392|3436|
> |0.4|38|621|260|3436|
> |0.5|47|132|147|3436|
>
> **Table 2b: Coreset performance (computation time) on Synthetic Data 2 ($N=200$, $T_i=1250$).**
>
> |$\varepsilon$|$T_S+T_C$(s) of **CRGMM**|$T_S+T_C$(s) of **Uni**|$T_S+T_C$(s) of **LFKF**|$T_\mathcal{X}$|
> | ------------- | ------------------------- | ----------------------- | ------------------------ | :--- |
> |0.1|694|1859|1687|9787|
> |0.2|139|1991|1484|9787|
> |0.3|51|527|832|9787|
> |0.4|43|408|450|9787|
> |0.5|57|389|277|9787|

---

> > ### Comment · Reviewer_xadL · 2021-08-23
> > **after rebuttal**
> >
> > Dear authors,
> >
> > 1. I have increased my score mainly due to your response regarding motivation of the model based time-series clustering. I strongly suggest adding the Frühwirth-Schnatter reference and also briefly discuss and reference other lines of time-series clustering, e.g. the NeurIPS'19 paper by S. Meintrup et al. but also the others suggested in my review. Also consider addressing the limitation to $AR(1)$ instead of $AR(p), p>1$. Note that the special case **and** the extension is discussed by Frühwirth-Schnatter (see below (4) and (5) in this reference).
> >
> > 2. I still find that this submission is **very incremental** extending [23] with the **exact methods** of [34]. Maybe it was not clear in my review that my **main concern** was not that [23] needs extensions to work for time series, but the extensions of this well-known work are **exactly the same** as in [34], which strongly limits novelty claims.
> >
> > 3. Since the motivation of the model is resolved in (1.) I strongly suggest to add some **real-world** dataset. The reason is that data generated from a model is usually very simple to fit or approximate within the same model, but real data might deviate more or less strongly from the model.
> >
> > Thanks for adding the experiment, I think the cam ready version will have additional space for a real-world example and there will be unlimited space in the appendix which can be referenced from the main body.

---

### Official Review · Reviewer_Wpms · 2021-07-16

**Rating:** 7
**Confidence:** 3

**Summary:**

In this paper the authors provide the definition of a coreset for time series clustering for entities generated from a gaussian mixture model and then build a coreset for the same problem.  The coreset is independent of both the number of entities N and the the number of Time stamps T_i for each entity i. They demonstarte the accuracy and efficiency of their coreset by superior empirical perfromance over synthetic data as compared to uniform sampling.

**Limitations And Societal Impact:**

The authors have mentioned their restriction to the GMM with autocorrelation as generative model for the time series data as a limitation.
However the main issue I have is that the experiments are only performed on synthetic data. It would be good if we can have some experiments with real datasets.
Also the only baseline used is uniform sampling. My question is, since the authors address the first part of coreset building procedure as a k-means clustering problem, is it also possible to atleast sample points in the frst part using some existing k-means coreset building procedure and compare with that?

I do not see any major negative societal potential impact of this work.

**Main Review:**

In this paper the authors provide coreset for time series data generated from GMM with autocorrelations.

Significance:
 Clustering time series data has become an important problem in ML due to huge amounts of data available over time. For scalabiltiy of clustering, coresets provide a very good option. From this view point this is an interesting and important paper. The theoretical results will be insightful to the coreset community in particular and the overall results are interesting for the ML community in general.

Originality:
To the best of my knowledge, this is the first paper on coreset for time series data in the unsupervised setting. The defintion of coreset provided by the authors for GMM time series clustering is different from traditional coreset in sense that along with a tuple of entity and time stamp it has two weight functions : one for the entity other for time.
Though the paper borrows heavily from existing coreset literature for e.g: 1)Feldman Langberg framework for constructing coresets, 2)the idea of introducing f' for the GMM clustering from [23][25] 3)the assumptions from [52,42,35,34] etc. ; I believe the application of these ideas to the setting of time series clustering for e.g bounding the sensitivities, pseudodimension etc. is non trivial. Overall I believe the paper has reasonable novelty.


Quality and  Clarity:
I have not checked the proofs in detail however the proofs look alright.
However I think the writing specifically the mathematical writing can be improved. There are a lot of variables and symbols used in the paper. It would be useful if some convention is followed for writing these variables. For e.g. vectors in small bold letters, matrices in capital bold letters, scalars in small letters etc. In current state it becomes very difficult while reading and one has to go to and fro to check if a given variable is vector, scalar etc.
I think there are minor typos at the following places: On line 183 eq. 2 should it not be f'(alpha(theta), theta) and not just alpha as a parameter.?
What is P on line 174?
What is tau in the input of algorithm 1 as subscripst of S?

The experiments show very positive results however they are only for synthetic data

Overall I feel this paper has merits that outweigh the flaws and is a good candidate for publication.




**Time Spent Reviewing:**

6

---

> ### Author Response · Authors · 2021-08-11
> **Thank you for your positive review and for your comments on the writing and presentation of empirical results. We accept your suggestions and provide clarifications for your queries. Further, in response to your suggestion, we have added a new benchmark using coreset for clustering cross-sectional data and show the empirical value of our time series clustering coreset. We hope these changes further strengthen your support for our paper. Please see below the point-by-point response.**
>
> > **"Quality and Clarity. It would be useful if some convention is followed for writing these variables. For e.g. vectors in small bold letters, matrices in capital bold letters, scalars in small letters etc. In current state it becomes very difficult while reading and one has to go to and fro to check if a given variable is vector, scalar etc."**
>
> *Response:* Thank you for the suggestion. We will revise the paper to follow the suggested convention for different types of variables.
>
> > **"On line 183 eq. 2 should it be $f'(\alpha(\theta), \theta)$ and not just alpha as a parameter?"**
>
> Yes, it should be $f'(\alpha’(\theta), \theta)$.
>
> > **"What is P on line 174? What is tau in the input of algorithm 1 as subscripst of S?"**
>
> *Response:* $P$ should be $P_\mathcal{X}$. There should be no $\tau$, it is a typo.  $\mathcal{S}_\tau^d$ should be $\mathcal{S}^d$.
>
> > **"They are only for synthetic data. It would be good if we can have some experiments with real datasets."**
>
> *Response:*  It would indeed be a valuable future direction to empirically evaluate our findings on real-world datasets. But given space constraints and the objective of our current paper to show the value of coresets for time series clustering, we chose to follow your suggestion below to add an additional benchmark using coresets for cross-sectional data (specifically [23]). We discuss the details of those results below. We hope you agree with our prioritization.
> .
>
> > **"Is it also possible to at least sample points in the first part using some existing k-means coreset building procedure and compare with that?"**
>
> *Response:* Thanks for your comments. In response to your suggestion,  in addition to uniform sampling, we also consider coreset based sampling for cross-sectional data (i.e., $T$=1, where entity-time pairs are considered independent) as a baseline. The results show that our coreset for time series clustering outperforms the coreset for cross-section clustering in [23], both in terms of fit ($\gamma_S$) and computation time. We hope this further clarifies the empirical value of the time series clustering coreset relative  to the existing literature, which only has developed clustering coresets for cross-sectional data.
>
>
> **Table 1a: Coreset performance (fit) on Synthetic Data 1 ($N=T=500$).**
>
> | $\varepsilon$ | $V_S^\star$ of **CRGMM** | $V_S^\star$ of **Uni** | $V_S^\star$ of **LFKF** | $V^\star$ | $\gamma_S^\star$ of **CRGMM** | $\gamma_S^\star$ of **Uni** | $\gamma_S^\star$ of **LFKF** |   size  |
> | ------------- | ------------------------ | ---------------------- | ----------------------- | --------- | ----------------------------- | --------------------------- | ---------------------------- | --- |
> | 0.1           |      2050                    |        2058                |            2069             |   2041  |  22   |   34  |  56  |  1514
> |   0.2       |     2093     |     2210     |   2264  |  2041   |   104  |   342  |   446  | 404
> |   0.3       |     2194     |     2398     |  2384   |  2041   |   306  |   714  |  686   | 191
> |   0.4       |     2335     |      3963    |   2705  |   2041  |   588  |   3844  |  1328   | 93
> | 0.5     |   2383   |   3304   |  3461   |  2041   |   684  |   2526  |   2840  | 72
>
> **Table 1b: Coreset performance (fit) on Synthetic Data 2 ($N=200, T=1250$).**
>
> | $\varepsilon$ | $V_S^\star$ of **CRGMM** | $V_S^\star$ of **Uni** | $V_S^\star$ of **LFKF** | $V^\star$ | $\gamma_S^\star$ of **CRGMM** | $\gamma_S^\star$ of **Uni** | $\gamma_S^\star$ of **LFKF** |  size   |
> | ------------- | ------------------------ | ---------------------- | ----------------------- | --------- | ----------------------------- | --------------------------- | ---------------------------- | --- |
> | 0.1           |         811                |       825                 |              841           |     812      |   2                            |        26                     |                 58             |   1718  |
> | 0.2           |          824                |       895                 |               871          |      812     |                24               |          166                   |             118                 |   447  |
> | 0.3           |           864               |      958                  |           994              |     812      |                  104             |           292                  |             364                 |   199  |
> | 0.4           |       860                   |      1190                  |              1114           |      812     |               96                |          756                   |               604               |   98  |
> | 0.5           |          910                |        1361                |            1284             |      812     |               196                |           1098                  |               944               |  71   |
>
>
> **Table 2a: Coreset performance (computation time) on Synthetic Data 1 ($N=T_i=500$).**
>
> | $\varepsilon$ | $T_S+T_C$(s) of **CRGMM** | $T_S+T_C$(s) of **Uni** | $T_S+T_C$(s) of **LFKF** | $T_\mathcal{X}$ |
> | ------------- | ------------------------- | ----------------------- | ------------------------ | :--- |
> | 0.1           |           109                |     2416                    |             1355             |   3436  |
> | 0.2           |            41               |    1054                     |             652             |  3436   |
> | 0.3           |            62               |     419                    |                392          |   3436  |
> | 0.4           |             38              |    621                     |         260                 |   3436  |
> | 0.5           |          47                 |    132                     |              147            |  3436   |
>
> **Table 2b: Coreset performance (computation time) on Synthetic Data 2 ($N=200$, $T_i=1250$).**
>
> | $\varepsilon$ | $T_S+T_C$(s) of **CRGMM** | $T_S+T_C$(s) of **Uni** | $T_S+T_C$(s) of **LFKF** | $T_\mathcal{X}$ |
> | ------------- | ------------------------- | ----------------------- | ------------------------ | :--- |
> | 0.1           |         694                  |     1859                    |             1687             |   9787  |
> | 0.2           |            139               |     1991                    |             1484             |   9787  |
> | 0.3           |             51              |     527                    |               832           |   9787  |
> | 0.4           |             43              |    408                     |                 450         |  9787   |
> | 0.5           |              57             |     389                    |              277            |  9787   |

---

> > ### Comment · Reviewer_Wpms · 2021-08-25
> > **After Rebuttal**
> >
> > Thanks for your detailed response and additional experiments. I maintain my score: 7 .

---

### Decision · Program_Chairs · 2021-09-27

**Decision:**

Accept (Spotlight)

**Comment:**


The paper suggests a coreset for clustering in the context of signals.
Indeed, there are too few coresets in this very active area.
There were concern regarding experimental results and novelty that the authors should address in the final version.
In addition, there are few coresets for signals that the authors should cite in the final version:

https://papers.nips.cc/paper/5581-coresets-for-k-segmentation-of-streaming-data
https://dl.acm.org/doi/abs/10.1145/2814569